# Timely Clinical Diagnosis through Active Test Selection

**Silas Ruhrberg Estévez**
University of Cambridge
Cambridge, UK
sr933@cam.ac.uk

**Nicolás Astorga**
University of Cambridge
Cambridge, UK
nja46@cam.ac.uk

**Mihaela van der Schaar**
University of Cambridge
Cambridge, UK
mv472@cam.ac.uk

## Abstract

There is growing interest in using machine learning (ML) to support clinical diagnosis, but most approaches rely on static, fully observed datasets and fail to reflect the sequential, resource-aware reasoning clinicians use in practice. Diagnosis remains complex and error prone, especially in high-pressure or resource-limited settings, underscoring the need for frameworks that help clinicians make timely and cost-effective decisions. We propose ACTMED (Adaptive Clinical Test selection via Model-based Experimental Design), a diagnostic framework that integrates Bayesian Experimental Design (BED) with large language models (LLMs) to better emulate real-world diagnostic reasoning. At each step, ACTMED selects the test expected to yield the greatest reduction in diagnostic uncertainty for a given patient. LLMs act as flexible simulators, generating plausible patient state distributions and supporting belief updates without requiring structured, task-specific training data. Clinicians can remain in the loop; reviewing test suggestions, interpreting intermediate outputs, and applying clinical judgment throughout. We evaluate ACTMED on real-world datasets and show it can optimize test selection to improve diagnostic accuracy, interpretability, and resource use. This represents a step toward transparent, adaptive, and clinician-aligned diagnostic systems that generalize across settings with reduced reliance on domain-specific data.

## 1 Introduction

Clinical diagnosis is a fundamental step in modern medical practice [1], providing the framework for future investigations and guiding treatment decisions, often determining patient outcomes. Yet it remains complex and error-prone, especially in fast-paced or resource-limited settings [2], where delays, misdiagnoses, and over-testing pose persistent global challenges [3], [4]. Additionally, the WHO projects a global shortage of more than 12 million qualified health professionals by 2035 [5]. Machine learning (ML) has emerged as a promising tool to support clinicians by improving diagnostic accuracy, optimizing test selection, and enabling earlier disease detection [6]–[9]. However, many current ML models operate under unrealistic assumptions, such as complete data availability [10], and fail to reflect the iterative, context-aware decision-making used by human clinicians [11].

**Clinical diagnosis.** Clinical diagnosis has traditionally followed local or national guidelines based on clinical trials and expert consensus [12], [13]. Although such guidelines improve outcomes by standardizing care, they are population-based and often inefficient at the individual level, with an estimated 40 to 60% of diagnostic tests being unnecessary [14]. Resource constraints further hinder access; for example, around 15% of clinician-ordered genetic tests go unperformed due to financial barriers [15]. In response, machine learning (ML) models have been proposed to support more personalized and efficient test ordering [16]. However, for these models to gain trust and adoption, they must be transparent and aligned with clinicians' reasoning processes [17], [18].

39th Conference on Neural Information Processing Systems (NeurIPS 2025).

**Current models for diagnosis.** The diagnostic process is inherently sequential, involving stepwise information gathering from patient examinations and tests [19]. Clinicians aim to achieve accurate early diagnoses while minimizing diagnostic costs, as timely intervention can significantly improve outcomes [20]–[22]. Prior models for early diagnosis often target a single disease and rely on specific modalities such as blood tests or imaging [23]–[25], which typically require large training data sets and assume full availability of modality [26]. In reality, this assumption rarely holds, and while imputation methods can handle missing data, they often introduce bias [10]. Additionally, many models treat diagnosis as a static classification task, overlooking the inherently dynamic and progressive nature of disease development and clinical reasoning. State-space models have been developed to capture disease trajectories [27]–[29], but they also require extensive retraining and struggle with balancing information acquisition costs [30]. Consequently, there remains a gap for generalizable frameworks that can reason dynamically under uncertainty and resource constraints [22], [31].

**LLMs for clinical diagnosis.** Recent advances in large language models (LLMs) have sparked interest in their use as general-purpose tools for medical decision-making [32], [33]. LLMs perform well on medical licensing exams [18], [34], [35] and are particularly effective in zero-shot settings [36], [37]. However, their direct deployment in clinical contexts faces challenges, including limited transparency and interpretability [38]. While chain-of-thought prompting improves interpretability [39], LLMs often deviate from the probabilistic optimum, although fine-tuning can improve their probabilistic reasoning [40]. Furthermore, recent work shows that LLM explanations often do not reflect their true internal reasoning processes, raising concerns about the faithfulness of chain-of-thought outputs [41]. It has also been shown that LLMs can approximate structured decision-making tasks, such as Bayesian optimization or decision tree induction, by leveraging latent inductive biases learned from large-scale text corpora [42]–[44]. Additionally, shifting LLM reasoning to the natural language solution space has been shown to enhance decision quality [45].

> **Contributions.** ① We motivate and formalize a transparent, stepwise diagnostic framework that aligns with clinical reasoning. ② We propose `ACTMED`, a probabilistic approach to timely diagnosis that uses Bayesian Experimental Design with LLMs to adaptively select tests based on their expected diagnostic utility. ③ We show that shifting reasoning from the LLM to the natural language output space can improve clinical decision-making. ④ We validate `ACTMED` on real-world datasets, demonstrating its ability to optimize test selection and improve diagnostic accuracy, interpretability, and resource use. This framework ultimately contributes to more transparent, adaptive, and clinician-aligned diagnostic processes.

## 2 Problem formalism

**Agent-based diagnosis model.** Let $T = \{1, 2, \ldots, T_{\max}\}$ denote the discrete time horizon representing stages in the decision process. The space of natural language is denoted by $\Sigma$, and $\mathcal{S} \in \Sigma$ represents the natural language instructions provided to the agent at each stage. The agent must return a diagnosis $d_t \in \mathcal{D}_t \subset \Sigma$, where $\mathcal{D}_t$ is the set of possible diagnoses at time $t$. We assume that a patient may present with a subset of all possible diagnoses $\mathcal{D}_{\text{true},t} \subset \mathcal{D}_t$. The agent independently estimates the posterior probability $P(y_{d_t} = 1 \mid K_t)$ for each diagnosis $d_t \in \mathcal{D}_t$, where $K_t$ is the information available at time $t$, and the belief is updated dynamically as new information is acquired.

At each time step $t \in T$, the agent observes a subset $K_t \subset \mathcal{X}_t$ of ground truth information $\mathcal{X}_t \subset \Sigma$, and may request additional information $U'_t \in \mathcal{U}_t$ from an external source, with $\mathcal{U}_t = \mathcal{X}_t \setminus K_t$. Here, $U'_t$ denotes the *random variable* corresponding to the requested test, while its realized outcome is written $u'_t$. The agent's objective is to minimize diagnostic error while penalizing costly information acquisition through a joint Lagrangian objective:

$$\mathcal{L}(y, u_t, \lambda) = \sum_t \mathbb{I}[y \neq y_{d_t}] + \lambda \sum_t c(u_t), \tag{1}$$

where $c(u_t)$ denotes the cost of the requested information and $\lambda$ is a weighting parameter controlling the trade-off between diagnostic accuracy and test cost.

**Optimal diagnostic test selection.** The challenge of optimal diagnostic test selection within Bayesian Experimental Design (BED) is to identify the test, $U_t^{(i)}$, that provides the greatest informa-

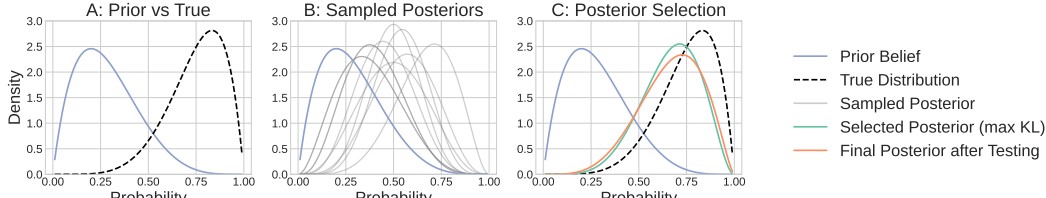

Figure 1: *Illustration of KL divergence-based diagnostic test selection.* **(A)** The agent begins with an initial *prior belief* about the likelihood of each diagnosis (blue), which represents current uncertainty, and seeks to approximate the unobserved *true posterior* (black dashed). **(B)** For each candidate diagnostic test, the agent uses a surrogate model to simulate possible *posterior distributions* (gray) conditioned on hypothetical outcomes, and computes the Kullback–Leibler (KL) divergence between each simulated posterior and the prior. **(C)** The test whose expected KL divergence is greatest (green) is selected, as it is expected to yield the largest information gain. After the test result is observed, the agent updates its belief to a new posterior distribution (orange).

---

tional utility regarding a diagnostic label, $y_{d_t}$. We consider a binary classification (e.g., sick/not sick), though this framework extends to multiple diagnoses by independent simultaneous application. The core objective is to select tests that maximally reduce epistemic uncertainty, the uncertainty stemming from our limited knowledge or model imperfections, which is reducible with new data, as opposed to aleatoric uncertainty, which is inherent system randomness.

To model the impact of potential information $U_t^{(i)}$, we employ a surrogate model to draw $M$ hypothetical realizations $u_t^{(i,j)} \sim P(U_t^{(i)})$. For binary classification, we assume the posterior distribution $P(y_{d_t} = 1 \mid K_t, u_t^{(i,j)})$ follows a Bernoulli distribution $\mathbb{B}(p_i)$, where $p_i \in [0, 1]$ is the success probability. In clinical practice, a test's value lies not just in reducing uncertainty, but in meaningfully shifting the probability of disease presence, especially across decision thresholds relevant to treatment decisions. While entropy-based formulations can be used, they may sometimes prioritize tests that reinforce confident but incorrect predictions. For instance, when a patient is initially assigned a very low disease probability, a truly informative test may increase this belief substantially. However, a test with no real diagnostic value might keep the prediction near zero, deceptively minimizing entropy (see Appendix B).

The information gain can also be expressed as the difference in KL divergence between the posterior and prior distributions of $y_{d_t}$. Maximizing this expectation ensures the selection of tests whose outcomes, on average, induce the most significant and diagnostically meaningful shifts in belief. This aligns with the core BED principle of maximizing information gain and directly addresses the clinical need to understand how a test will alter diagnostic probabilities, especially concerning critical decision thresholds. Given the unknown nature of these prior and posterior distributions, we utilize our surrogate model to generate samples $j$ representing hypothetical test results. Let $p_{\text{prior}}$ represent the prior, and $p_{\text{post}}$ the posterior probability distribution: $p_{\text{prior}}^{(j)} \sim P(y_{d_t} = 1 \mid K_t)$, $p_{\text{post}}^{(j)} \sim P(y_{d_t} = 1 \mid K_t, u_t^{(i,j)})$. The expected KL divergence is then computed as the average KL divergence over the $M$ samples from both distributions:

$$\mathbb{E}[\text{KL}(\mathbb{B}(p_{\text{post}}) \parallel \mathbb{B}(p_{\text{prior}}))] = \frac{1}{M} \sum_{j=1}^{M} p_{\text{post}}^{(j)} \log \left( \frac{p_{\text{post}}^{(j)}}{p_{\text{prior}}^{(j)}} \right) + \left( 1 - p_{\text{post}}^{(j)} \right) \log \left( \frac{1 - p_{\text{post}}^{(j)}}{1 - p_{\text{prior}}^{(j)}} \right). \quad (2)$$

The optimal test $U_t^{(i^*)}$ is the one that maximizes this expected KL divergence, i.e. $i^* = \arg\max_i \mathbb{E}[\text{KL}(\mathbb{B}(p_{\text{post}}) \parallel \mathbb{B}(p_{\text{prior}}))]$. The process of KL-guided diagnostic test selection and belief updating is illustrated in Figure 1. A full derivation is included in Appendix B.

> **Example 1:** An agent is tasked with diagnosing chronic kidney disease (CKD), aiming to establish the correct diagnosis $d_t$ as early as possible while minimizing additional diagnostic evaluations $U_t^{(i)}$ due to budget constraints and test delays. At each time point $t$, the agent has access to clinical information $K_t$, including demographics and previous test results, and can request further information $U_t^{(i)}$, such as lab tests or imaging, to refine the diagnosis $d_t$.

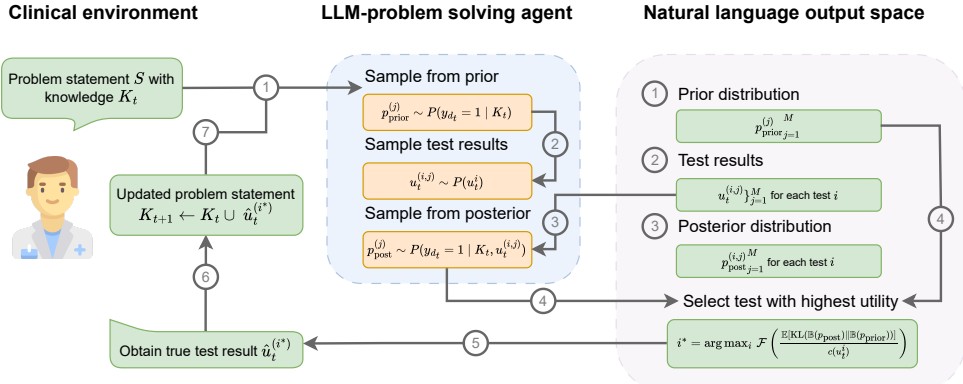

Figure 2: *Overview of* `ACTMED`. A clinician queries the system (Step 1), prompting the agent to estimate prior disease risk. The agent simulates test outcomes (Step 2), updates beliefs (Step 3), and computes expected KL divergence to select the most informative test (Step 4). The clinician reviews and conducts the test (Steps 5–6), updates the context (Step 7), and the process repeats iteratively.

## 3   `ACTMED`: Probabilistic reasoning for clinical diagnosis

**Surrogate model sampling.**   `ACTMED` estimates the expected utility of potential diagnostic tests by evaluating their hypothetical impact on posterior diagnostic beliefs. At each time step, for every candidate test $U_t^{(i)} \in \mathcal{U}_t$, the agent draws a set of plausible outcomes $u_t^{(i,j)} \sim P(U_t^{(i)} \mid K_t)$, where $K_t$ denotes the current patient knowledge base. Each hypothetical outcome $u_t^{(i,j)}$ induces a posterior probability $P(y_{d_t} = 1 \mid K_t, u_t^{(i,j)})$, allowing estimation of the expected information gain associated with performing that test.

This process requires a reliable mechanism for approximating the conditional outcome distribution $P(U_t^{(i)} \mid K_t)$—that is, for predicting what values a test might realistically produce given the patient's current profile. In well-characterized physical systems, such distributions can be modeled mechanistically, and Bayesian Experimental Design (BED) has been successfully applied in domains such as engineering [46], physics [47], and neuroscience [48]. In clinical and biomedical contexts, however, the underlying dynamics are rarely known in closed form: interactions among physiological, biochemical, and behavioral variables are nonlinear, high-dimensional, and only partially observable [49]. This renders direct physical modeling infeasible and motivates the use of data-driven surrogates capable of capturing complex empirical relationships between tests and patient states.

**LLM-driven sampling for `ACTMED`.**   To overcome the absence of explicit mechanistic models, we introduce Large Language Models (LLMs) as flexible generative surrogates that encode prior medical knowledge learned from large-scale clinical and biomedical corpora. When prompted with structured patient information $K_t$, an LLM acts as a conditional sampler from an implicit joint distribution over diagnostic variables. In this setup, the model generates plausible realizations $u_t^{(i,j)}$ for unobserved tests $U_t^{(i)}$, effectively approximating $P(U_t^{(i)} \mid K_t)$ without requiring explicit mechanistic equations or task-specific fine-tuning. These generated samples allow `ACTMED` to simulate how new evidence would alter diagnostic beliefs and, consequently, to compute the expected information gain before any real test is performed. Prior studies have demonstrated that LLMs can anticipate clinical test outcomes and model disease trajectories with high fidelity [50], suggesting that they encode inductive biases suitable for probabilistic reasoning in diagnostic decision-making. Figure 2 illustrates how `ACTMED` supports clinician-driven diagnostic reasoning.

We quantify the diagnostic value of each candidate test using an information-theoretic cost–benefit criterion. Let $I(U_t^{(i)}) := \mathbb{E}[\mathrm{KL}(\mathbb{B}(p_\mathrm{post}) \parallel \mathbb{B}(p_\mathrm{prior}))]$ denote the expected information gain obtained from performing test $U_t^{(i)}$. To account for practical constraints such as test invasiveness, financial cost, and time delay, we include a cost term $c(U_t^{(i)})$ in the decision objective. Relaxing the Lagrangian formulation in Eq. (1) and assuming logarithmic scaling of cost, $c(U_t^{(i)}) = \log c^*(U_t^{(i)})$, we define

the normalized utility:

$$\mathcal{F}(U_t^{(i)}) = \frac{I(U_t^{(i)})}{c(U_t^{(i)})}.$$

This formulation prioritizes tests that maximally reduce epistemic uncertainty relative to their expected cost, enabling efficient, interpretable, and resource-aware diagnostic reasoning.

---

ACTMED selects the next diagnostic test at each time step $t$ as follows:

1. Initialize the prior belief $p_{\text{prior}} = P(y_{d_t} = 1 \mid K_t)$ based on current knowledge $K_t$.

2. For each candidate test $U_t^{(i)}$, sample $M$ possible outcomes $\{u_t^{(i,j)}\}_{j=1}^M \sim P(U_t^{(i)})$.

3. Compute posterior beliefs $p_{\text{post}}^{(i,j)} = P(y_{d_t} = 1 \mid K_t, u_t^{(i,j)})$.

4. Estimate expected information gain $\mathbb{E}[\text{KL}(\mathbb{B}(p_{\text{post}}) \parallel \mathbb{B}(p_{\text{prior}}))]$ and define utility $\mathcal{F}(U_t^{(i)})$.

5. Select the most informative test $U_t^{(i^*)} = \arg\max_i \mathcal{F}(U_t^{(i)})$ and observe outcome $\hat{u}_t^{(i^*)}$.

6. Update the knowledge base: $K_{t+1} \leftarrow K_t \cup \{\hat{u}_t^{(i^*)}\}$.

---

**Deciding when to stop information acquisition.** Our KL divergence-based criterion naturally supports adaptive test acquisition by recommending a new test only when it is expected to significantly update the current belief. At each step, the agent maintains a disease belief $p_{\text{prior}} \in [0, 1]$. Diagnosis proceeds until this belief is sufficiently confident, measured relative to a decision threshold $\theta = 0.5$, regardless of the expected results of any further tests. We define the confidence gap as $\delta = |p_{\text{prior}} - \theta|$, and set a target posterior belief $q_{\text{target}} = \theta \pm \gamma\delta$, where $0 \leq \gamma \leq 1$ is a hyperparameter that controls the desired confidence margin before stopping. The sign is chosen to move the target posterior toward the decision boundary $\theta$; the hyperparameter $\gamma$ controls how much of that distance is required to justify acquiring the test. A test is acquired only if the expected KL divergence from the current belief to the candidate posterior satisfies: $\mathbb{E}[\text{KL}(\mathbb{B}(p_{\text{post}}) \parallel \mathbb{B}(p_{\text{prior}}))] \geq \mathbb{E}[\text{KL}(\mathbb{B}(q_{\text{target}}) \parallel \mathbb{B}(p_{\text{prior}}))]$. Further tests are acquired only if at least one remaining test is expected to meaningfully shift the current belief; otherwise, the model is sufficiently confident that no additional information, regardless of outcome, would alter the prediction.

**Mitigating LLM hallucinations.** We utilize LLMs as surrogate models for BED, employing structured prompts with three components: ① **Context specification**: A brief description of the clinical scenario and disease. ② **Known information** $(K_t)$: Clinical observations and test results available at time $t$ formatted in a clinical vignette (see Appendix C). ③ **Task-specific instruction**: Directives for the model's output, such as predicting test outcomes $u_t^{(i,j)} \sim P(u_t^i)$, diagnosis probabilities $P(y_{d_t} = 1 \mid K_t)$, or selecting the most informative next test. Full prompt examples are given in Appendix D.

**Encouraging diverse test result sampling.** We enhance model robustness and capture diagnostic uncertainty with three strategies: ① **Avoiding population averages**: The model is prompted to sample from a broader distribution of possible outcomes. ② **Increased sampling temperature**: This introduces greater randomness, reflecting higher uncertainty and improving prediction diversity. ③ **Sampling both disease presence and absence**: The model is instructed to sample outcomes under both conditions to ensure balanced and varied predictions.

## 4 Experiments

We evaluate two central hypotheses derived from the discussions in the previous sections:

- **H1)** Large Language Models (LLMs) can accurately approximate the distributions of diagnostic test outcomes based on available patient information.

- **H2)** Incorporating LLM-based Bayesian Experimental Design (BED) enables more accurate and efficient diagnosis through adaptive, information-driven test selection.

To examine these hypotheses, we design experiments spanning three progressively challenging diagnostic tasks, each corresponding to a distinct and clinically relevant disease context.

**Chronic Kidney Disease (CKD)** affects over 700 million people worldwide and is responsible for more than 3 million deaths annually. Early detection is essential for slowing disease progression and preventing renal failure [51]. **Hepatitis C** infects approximately 57 million people and causes around 300,000 deaths each year. Its

diagnosis is often delayed because it relies on specialized confirmatory tests, underscoring the need for surrogate blood-based indicators [52], [53]. **Diabetes**, which affects roughly 500 million individuals globally, remains a leading cause of cardiovascular disease and premature mortality. These three single-condition datasets provide controlled environments to assess `ACTMED`'s diagnostic reasoning, and test-selection efficiency across patient cohorts. They allow fine-grained comparison of how well LLM-driven surrogate sampling aligns with empirical data and whether adaptive BED reasoning leads to more efficient diagnostic strategies.

To evaluate generalization beyond these controlled settings, we further introduce a custom OSCE-style (Objective Structured Clinical Examination) dataset derived from *AgentClinic* [54]. This dataset comprises diverse, multi-condition clinical cases designed to mimic real-world diagnostic encounters, where the agent must reason across varying clinical contexts. It serves as a robust stress test for zero-shot diagnostic generalization, assessing whether the proposed framework can transfer its reasoning beyond narrowly defined single-disease tasks.

## 4.1 LLMs accurately predict distributions of diagnostic tests

**Surrogate sampling evaluation.** The fidelity of surrogate samples generated by LLMs is critical for the reliability of our Bayesian diagnostic framework. For each patient, the model was tasked with sampling ten plausible outcomes for every missing laboratory feature, conditioned on the available clinical context. The resulting synthetic feature distributions were then compared to the empirical distributions observed in the real datasets. Table 1 reports normalized Wasserstein and Energy distances between LLM-generated and empirical feature distributions, providing quantitative measures of distributional alignment. Across all datasets, GPT-4o achieved the lowest average distances, indicating strong agreement

Table 1: *Normalized distribution matching metrics (lower is better).* Average Wasserstein and Energy Distance (mean ± std).

| Dataset | Model | Wasserstein | Energy |
|---------|-------|-------------|--------|
| Diabetes | GPT-4o | 0.110 ± 0.038 | 0.256 ± 0.079 |
| | GPT-4o-mini | 0.117 ± 0.046 | 0.269 ± 0.094 |
| Hepatitis | GPT-4o | 0.130 ± 0.157 | 0.265 ± 0.191 |
| | GPT-4o-mini | 0.173 ± 0.237 | 0.330 ± 0.254 |
| Kidney | GPT-4o | 0.082 ± 0.055 | 0.203 ± 0.102 |
| | GPT-4o-mini | 0.082 ± 0.057 | 0.203 ± 0.104 |

with real-world feature distributions. GPT-4o-mini showed slightly higher divergence, consistent with its smaller model capacity. These results confirm that both models generate physiologically coherent and statistically realistic samples suitable for surrogate-based Bayesian Experimental Design. Feature-level deviations are visualized in Fig. 3, which illustrates per-test mean absolute error (MAE) across five random seeds, highlighting that most biomarkers are reproduced with high fidelity, with slightly higher errors observed for high-variance biochemical markers. The complete distributions produced during the sampling are shown in Appendix E.

## 4.2 LLM-based BED improves diagnostic accuracy and efficiency

**Timely diagnosis under resource constraints.** We evaluate diagnostic accuracy under the constraint of acquiring only three clinical tests per patient, simulating real-world limitations such as acquisition delays and resource costs. Although our framework can accommodate arbitrary test cost functions, we use uniform costs across tasks to maintain consistency, as defining task-specific costs would require expert clinical input. Importantly, `ACTMED` is model-agnostic: its performance depends on the quality of the underlying surrogate model rather than any specific LLM architecture. We validate this by applying it across models of varying capacity; GPT-4o-mini and GPT-4o. Details for all datasets are given in Appendix C. We also provide a detailed failure analysis of `ACTMED` when using less powerful models that do not produce accurate samples in Appendix E.

We benchmark `ACTMED` against the following baselines that use the same models and risk prediction prompts to show how our approach can improve the performance of specific models:

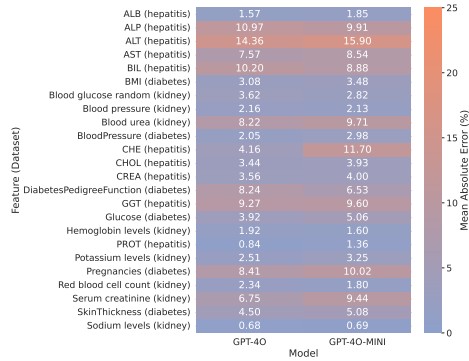

Figure 3: *Feature sampling performance.* Heatmap showing the best MAE percentage for each test averaged across 5 seeds.

- ➢ LLM classifier: No three-feature constraint; uses **all features** for direct classification.
- ➢ **Random** selection: Picks three features randomly as a stochastic baseline.
- ➢ **Global best** fixed subset: Selects a predefined set of three features prior to observing individual patient data for the task, mimicking diagnostic guidelines.
- ➢ **Implicit** selection: Selects three features actively using LLM reasoning at each step without Bayesian modelling.

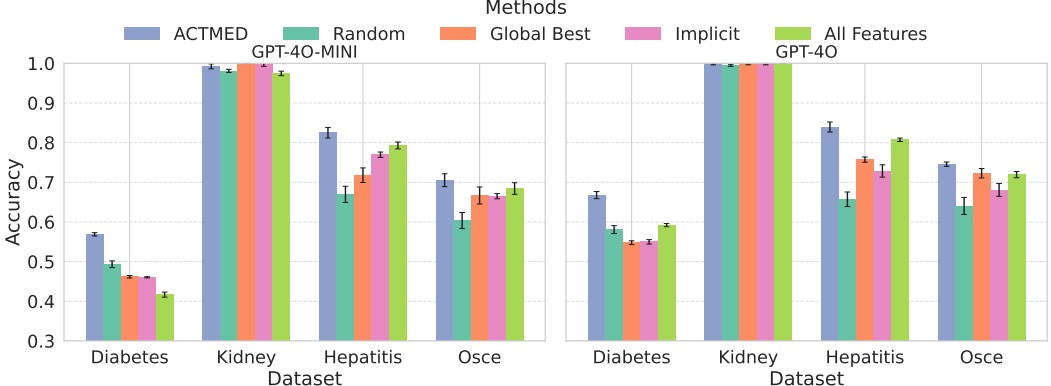

Figure 4: *Model accuracy across methods and datasets.* Bars show mean accuracy over five seeds; error bars indicate standard deviation.

**Chronic Kidney Disease (CKD).** Chronic kidney disease is typically diagnosed using biomarkers such as serum creatinine or glomerular filtration rate (GFR), with well-established clinical thresholds [51]. As a result, this represents a relatively straightforward classification task. Both models achieved near-perfect accuracy, even under feature selection constraints, indicating that LLMs possess strong prior knowledge of CKD's clinical presentation. This highlights their potential to support diagnostic decision-making in settings where key features are well understood. Consequently, we focus subsequent analysis on more challenging tasks, such as hepatitis and diabetes, where diagnostic ambiguity is higher and performance depends more heavily on effective test selection. Full performance metrics for all datasets and feature selection evaluation are provided in Appendix E.

**Hepatitis.** This task focuses on diagnosing hepatitis C using liver function tests [52]. Unlike CKD, hepatitis C often produces subtle and non-specific alterations in blood biomarkers, which can also be influenced by a range of other conditions. In our experiments, `ACTMED` consistently outperformed other feature selection methods across all model types (see Figure 4). Small improvements in GPT-4o compared to GPT-4o-mini are also noted. Interestingly, it even surpassed the full-information baseline, indicating that the targeted selection of informative features can improve diagnostic precision. This effect is visualized in Figure 5, which shows a clear increase in the average diagnostic accuracy across all datasets using `ACTMED` as additional features are acquired sequentially.

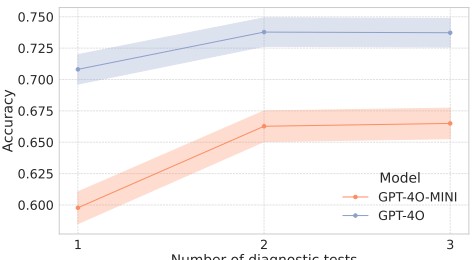

Figure 5: *Sequential diagnosis refinement.* Accuracy improves as additional tests are acquired; error bars show standard deviation across datasets.

**Diabetes.** Diagnosing diabetes is the most challenging of the three tasks, as no single definitive test is available in the datasets. Instead, diagnosis must be inferred by synthesizing multiple indirect indicators; such as blood glucose, body mass index (BMI), and blood pressure; to estimate overall disease risk [55]. Across both models, `ACTMED` consistently outperformed all other baselines, including the full-information setting. Performance further improved with the more capable GPT-4o model, highlighting the benefit of stronger underlying surrogate models. These results suggest that deliberate, targeted test acquisition not only enhances diagnostic accuracy but also improves generalization by reducing the influence of irrelevant or misleading features [56].

## 4.3 Implicit Selection Lacks Personalization

For the datasets where the baseline feature selection by the model performed notably worse than `ACTMED`, we further analysed the tests selected by each method by evaluating how frequently models chose features identified as globally optimal prior to observing any patient-specific data (see Figure 6). Random selection serves as a stochastic baseline. Across all model-dataset pairs, implicit selection methods showed a strong bias toward globally optimal features, significantly more so than `ACTMED`. This effect was especially pronounced in the diabetes dataset, where both models almost exclusively selected globally optimal features, despite `ACTMED` achieving higher predictive accuracy. These findings suggest that LLM-based implicit selection may struggle to capture patient-specific uncertainty and adaptively personalize test acquisition, relying instead on prior knowledge about general test utility. A more detailed feature selection analysis is performed in Appendix E.

## 4.4 KL-Based Stopping Minimizes Redundant Testing

`ACTMED` not only provides greater transparency than implicit feature selection methods by generating intermediate outputs and utilizing Bayesian test selection, but also introduces a principled stopping criterion for diagnostic testing based on the expected shift in the posterior distribution, measured by KL divergence. A representative example in Table 2 shows that KL divergence consistently decreases across selection rounds, naturally allowing the stopping criterion to trigger when the expected informational gain becomes negligible. We evaluated this criterion using thresholds $\gamma \in \{0.3, 0.5, 0.7\}$, which represent varying levels of evidence required to continue testing. Baselines based on implicitly selected features or global optima did not allow early stopping and had access to all three selected tests. Our KL-based stopping method achieved superior accuracy with significantly fewer diagnostic steps (see Figure 7). As expected, higher values of $\gamma$ reduce the number of

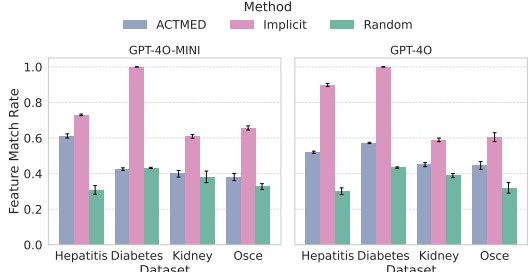

Figure 6: *Feature selection analysis.* Average selection frequency and variability across five random seeds are compared against the three globally optimal features. The implicit model consistently favours these global features with low variability, indicating limited personalization across patients.

diagnostic tests selected. At the conservative threshold ($\gamma = 0.6$), it reduced the average number of tests to under two across all datasets and models, except for the hepatitis dataset with GPT-4o. The method reduces overall diagnostic burden by nearly 50% while providing comparable or superior accuracy compared to baseline feature selection (see Table 3).

## 4.5 Clinician-in-the-loop evaluation

We conducted a clinician-in-the-loop evaluation involving three experienced clinicians and two senior medical students, assessing a total of 450 diagnostic test decisions (10 simulated diagnostic traces across the three datasets, each reviewed by five evaluators with three decisions per trace). Experts judged `ACTMED`'s test selections and resulting risk adjustments as clinically reasonable in $94.5 \pm 1.4\%$ of cases, underscoring its reliability as a decision-support tool. Clinicians emphasized that they would be reluctant to trust diagnostic suggestions from a purely black-box LLM without transparent reasoning. In contrast, `ACTMED`'s step-by-step decision process was repeatedly praised for fostering confidence in the system's reasoning.

Several clinicians remarked that `ACTMED`'s Bayesian decision framework closely reflects their own reasoning process when selecting diagnostic tests under uncertainty, particularly in situations lacking explicit clinical guidelines. They also noted instances where real-world practice departs from a purely Bayesian rationale—for example, when standardized protocols prescribe a specific test (such as PCR for COVID-19 diagnosis) or when certain screening procedures are routinely performed despite limited diagnostic yield (e.g., broad cancer panels). Nonetheless, participants emphasized that decision-support systems are most valuable in non-trivial clinical scenarios, where genuine uncertainty exists and no clear diagnostic pathway is defined.

Table 2: Representative feature selection rounds from the Hepatitis C dataset. Darker blue indicates higher KL divergence.

| Selection 1 | Selection 2 | Selection 3 |
|---|---|---|
| ALB: 0.066 | ALB: 0.014 | ALB: 0.016 |
| ALP: 0.041 | ALP: 0.028 | — |
| ALT: 0.172 | ALT: 0.014 | ALT: 0.019 |
| AST: 0.316 | — | — |
| BIL: 0.191 | BIL: 0.025 | BIL: 0.012 |
| CHE: 0.137 | CHE: 0.016 | CHE: 0.015 |
| CHOL: 0.124 | CHOL: 0.020 | CHOL: 0.016 |
| CREA: 0.084 | CREA: 0.021 | CREA: 0.016 |
| GGT: 0.138 | GGT: 0.006 | GGT: 0.018 |
| PROT: 0.126 | PROT: 0.020 | PROT: 0.020 |

## 4.6 OSCE-style evaluation

The three datasets discussed above each represent a single-diagnosis task. While this setting is well-suited to analyze `ACTMED`'s behaviour in controlled diagnostic contexts such as test selection, it does not fully capture the diversity of clinical reasoning required in practice. To evaluate `ACTMED`'s performance in a more realistic scenario, we constructed a **custom OSCE-style dataset** based on cases from the *AgentClinic* framework [54]. We selected **114 representative cases** with sufficiently rich test results and generated corresponding **synthetic negatives** by adjusting laboratory values toward physiological ranges that make the diagnosis unlikely. This allowed us to assess both diagnostic discrimination and generalization under realistic clinical variability. Across these multi-diagnosis settings, `ACTMED` continued to outperform baseline methods, maintaining superior calibration and diagnostic accuracy (see Figure 4). These results suggest that `ACTMED`'s sequential Bayesian reasoning scales effectively to diverse and complex diagnostic tasks, further supporting its potential for integration into clinical assessment frameworks such as OSCEs.

# 5 Discussion

**BED improves LLM-based clinical diagnosis.** ACTMED, integrating Bayesian Experimental Design (BED) decision-making with large language models (LLMs), outperforms naive LLM-based feature selection methods in accuracy, cost-awareness, personalization, and interpretability across several datasets. In the hepatitis and diabetes datasets it even surpasses the baseline model that uses all available features. While this may seem counter-intuitive, it aligns with traditional machine learning findings where feature selection reduces noise and over-fitting [57]. Implicit sequential and global selection strategies provide strong baselines but occasionally fail to identify the most informative features. We hypothesize this arises from the model's difficulty in accurately representing how test results influence diagnostic risk due to limited domain-specific data. By

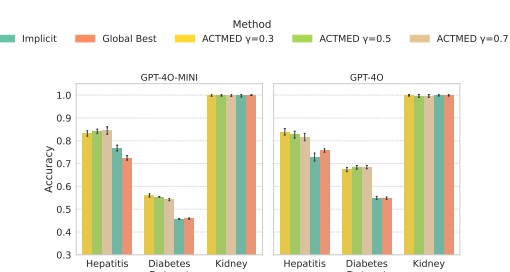

Figure 7: *KL-based early stopping criterion evaluation.* Bars represent the mean accuracy across 5 seeds, with error bars indicating standard deviation.

shifting reasoning from the input space to the solution space, our framework enables better probabilistic inference for test selection, improving decision-making under resource constraints and leveraging LLMs' strength in contextual prediction while compensating for their limitations in Bayesian reasoning.

**Comparisons to other diagnostic frameworks.** Conventional clinical diagnosis relies on rule-based, population-derived guidelines that generalize well but often lack personalization and may delay detection, especially in asymptomatic patients [58]. While machine learning (ML) and deep learning (DL) approaches promise earlier detection, they are typically task-specific, lack individualized reasoning, and offer limited transparency. Even interpretability techniques such as attention maps fall short in addressing the broader need for explainable and collaborative decision-making. Recent efforts have begun applying LLMs to clinical diagnosis, but naive implementations have proven inadequate [38], particularly in our study, where LLMs struggled with personalization and had limited ability to assess the effectiveness of candidate tests. Our framework addresses these challenges by leveraging the generative strengths of LLMs within a BED framework. This enables explicit reasoning under uncertainty, personalized diagnostic pathways, and principled test selection based on expected information gain. Importantly, it also supports transparency by incorporating clinicians in the process [39]. A summary comparison of capabilities is presented in Table 4.

**Clinical relevance.** Our framework's key advantage over standard end-to-end classifiers is its ability to involve clinicians directly in the diagnostic process. At each decision step, clinicians can review simulated test outcomes and assess their impact on diagnostic probabilities (see Appendix E). The framework issues test recommendations rather than fixed decisions, allowing clinicians to override suggestions based on context and re-query with alternative test results. While capable of

Table 3: **Average number of tests selected under KL-based termination for varying $\gamma$ values.** Higher $\gamma$ requires stronger evidence to continue testing. Results are reported as mean $\pm$ standard deviation across five random seeds.

| Model | $\gamma$ | Hepatitis | Diabetes | Kidney |
|---|---|---|---|---|
| GPT-4O-MINI | 0.3 | $2.36 \pm 0.68$ | $2.45 \pm 0.71$ | $1.60 \pm 0.75$ |
| | 0.5 | $1.87 \pm 0.77$ | $2.09 \pm 0.75$ | $1.48 \pm 0.68$ |
| | 0.7 | $1.48 \pm 0.73$ | $1.90 \pm 0.74$ | $1.28 \pm 0.52$ |
| GPT-4O | 0.3 | $2.67 \pm 0.60$ | $2.27 \pm 0.84$ | $1.35 \pm 0.64$ |
| | 0.5 | $2.41 \pm 0.67$ | $1.90 \pm 0.89$ | $1.09 \pm 0.35$ |
| | 0.7 | $2.27 \pm 0.65$ | $1.69 \pm 0.84$ | $1.04 \pm 0.24$ |

providing a final disease prediction, the model serves primarily as a decision-support tool, leaving diagnostic judgment to the clinician. By highlighting the most informative tests, it streamlines clinical reasoning and reduces cognitive burden. We emphasize that current LLMs are not yet ready for direct autonomous diagnosis but may be better suited as decision-support tools, assisting clinicians in structuring and refining the diagnostic process. Importantly, safety can be enhanced by constraining the model to recommend only tests that are clinically approved for the suspected conditions, ensuring alignment with established diagnostic pathways and regulatory guidelines.

**Limitations.** Our evaluation is currently constrained by dataset scale and scope. The combined sample size across all datasets is approximately 1,000 patients, with a limited number of covariates per condition. Each task is framed as a binary diagnostic decision, though the framework is applied across multiple distinct diseases rather than within a single multi-label setting. Extending ACTMED to larger, more heterogeneous datasets with richer feature spaces and overlapping comorbidities would better reflect the complexity of real-world clinical diagnosis.

Table 4: **Capability matrix for clinical reasoning approaches.** Only our proposed framework satisfies all key criteria for transparent, timely diagnosis under resource constraints.

| Method | Timely Diagnosis | Personalized Reasoning | Resource-Constrained | Zero-Shot Generalization | Unstructured Data | Transparent Explanations |
|---|---|---|---|---|---|---|
| Rule-Based Systems | ✗ | ✗ | ✓ | ✓ | ✗ | ✓ |
| Classical ML | ✓ | ✗ | ✗ | ✗ | ✗ | – |
| Deep Learning | ✓ | ✗ | ✗ | ✗ | ✗ | ✗ |
| Naïve LLM | ✓ | ✗ | ✗ | ✓ | ✓ | ✗ |
| **Ours** (ACTMED) | ✓ | ✓ | ✓ | ✓ | ✓ | ✓ |

Our framework is also currently limited to binary classification. Future work should extend evaluation to multi-label datasets and co-morbidities for broader clinical applicability. The diagnostic process considers only categorical and numerical features, with free-text outputs (e.g., imaging or pathology reports) not yet incorporated, though structured representations could be included. While LLMs can interpret free-text, this adds complexity beyond our current structured setting. Additionally, our method requires more LLM queries than simpler heuristics, which may pose challenges in resource-constrained environments. However, in high-stakes domains such as healthcare, the added computational cost is justified by improved decision-making, reduced uncertainty, and fewer unnecessary tests.

ACTMED also depends on strong, high-capacity LLMs to generate physiologically coherent feature distributions and uncertainty-aware predictions. Simpler models fail to reproduce accurate surrogate samples or stable posterior estimates, underscoring the current reliance on foundation-scale models. LLMs are often criticized for their black-box nature and susceptibility to hallucinations. While these issues persist, our framework mitigates them by generating interpretable intermediate outputs, such as sampled feature distributions and uncertainty-calibrated diagnostic probabilities, thereby improving transparency in the reasoning process. Biases in model behavior also remain a concern, particularly for under-represented patient subgroups. By explicitly exposing how test results influence posterior diagnostic beliefs, clinicians can better identify and assess such biases in real time, enabling oversight and corrective action when needed.

Clinical diagnosis generally follows a two-step process [19]: (1) an initial assessment, where the clinician formulates a primary diagnosis and a list of differential diagnoses, and (2) a targeted testing phase to confirm or refute these hypotheses. Our implementation of ACTMED addresses the second step by optimizing test selection across individual diseases, independent of how the differential list is generated. This one-vs-all formulation mirrors established clinical reasoning practices such as the Wells score for deep vein thrombosis [59] and the $CHA_2DS_2$-VASc score for atrial fibrillation-related stroke [60] where each disease likelihood is evaluated separately. While ACTMED can in principle be extended to sequentially handle multiple conditions, our present experiments focus exclusively on binary disease-specific tasks and do not include multi-label datasets.

**Conclusions and Impact.** We present ACTMED, a probabilistic framework for clinical diagnosis that uses sequential, information-theoretic decision-making to refine beliefs about disease states. Unlike traditional diagnostic models, our approach actively selects and interprets tests to inform decision-making, with LLMs serving as flexible surrogate simulators that predict test outcomes and update beliefs without task-specific training. The framework is model-agnostic: its performance depends on the fidelity of the surrogate model rather than any particular LLM architecture. Beyond improving diagnostic accuracy, ACTMED demonstrates how probabilistic reasoning and large-scale language models can be integrated to support data-efficient, uncertainty-aware clinical decision-making. As LLMs and generative models continue to advance, frameworks like ACTMED could enable clinician-in-the-loop systems that optimize diagnostic efficiency, improve interpretability, and mitigate resource constraints in healthcare. Future research should benchmark a range of model architectures and training paradigms to quantify their impact on diagnostic reasoning, extend the framework to multimodal and longitudinal data.

# 6 Acknowledgments and Disclosure of Funding

We thank the anonymous NeurIPS reviewers, members of the van der Schaar Lab, and Andrew Rashbass for their insightful comments and suggestions. N.A. gratefully acknowledges support from the W.D. Armstrong Trust. This work was supported by the Microsoft AI for Good Research Lab, including Azure sponsorship credits.

# 7 Reproducibility

The code and datasets to reproduce the main findings of this paper are available under https://github.com/Sr933/actmed.

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

# Supplementary Material

## A  Extended related works

**LLMs and Reinforcement Learning for Clinical Diagnosis.**  There has been growing interest in validating large language models (LLMs) on clinical diagnosis tasks, alongside efforts to fine-tune these models for biomedical datasets. Interestingly, recent work has shown that general-purpose LLMs can sometimes outperform specialized models even on domain-specific benchmarks [61]. To improve clinical reasoning, some approaches have involved training LLMs on medical dialogues [62], while others have incorporated structured knowledge sources such as medical knowledge graphs, particularly for tasks resembling medical licensing exams [63]. Beyond static evaluation, researchers have explored interactive diagnostic frameworks that allow LLMs to collaborate with clinicians. For instance, agentic networks of LLMs have been shown to enhance diagnostic reasoning compared to individual models, though their effectiveness strongly depends on the quality of the underlying language model [41]. Conversely, simply providing clinicians with access to unrefined LLMs has not yielded significant improvements in diagnostic performance [64].

Reinforcement learning (RL) has also been proposed as a way to teach models clinically grounded decision-making behaviours. Early work by Ling et al. demonstrated how deep reinforcement learning can be used to learn diagnostic reasoning directly from clinical narratives [65]. Subsequent studies have extended this paradigm to cost-sensitive and hierarchical frameworks, where the model learns to balance diagnostic accuracy against the cost of additional tests or procedures [66], [67]. Such approaches typically require substantial task-specific training and supervision, which can limit generalization to new diagnostic settings. In contrast, ACTMED combines Bayesian Experimental Design with LLM-based generative priors to simulate plausible test outcomes at inference time. This enables it to operate without any task-specific retraining, while still providing adaptive and interpretable test selection. We therefore view DRL-based diagnostic agents as complementary to our approach.

### Bayesian methods in medicine

Bayesian methods have long been influential in clinical research, particularly for optimizing trial design, such as adaptive patient allocation and dose-finding strategies in early-phase drug development [68], [69].  In medical imaging, Bayesian active learning approaches have been used to strategically acquire informative data points, improving model efficiency and performance [70].  More broadly, the diagnostic process itself is naturally aligned with Bayesian reasoning, where clinical evidence incrementally updates probabilistic beliefs about possible conditions [71]. In diagnostic settings, Bayesian approaches have been specifically applied to optimize test selection by leveraging the known sensitivity and specificity of various diagnostic tools [72]. Early foundational work in this domain illustrated the potential of step-wise Bayesian integration of test results, where each successive diagnostic query is strategically chosen based on its expected diagnostic value [73]. These methods effectively utilize probabilistic reasoning over potential test outcomes, exploiting the inherent accuracy profiles of available tests to make efficient decisions. Furthermore, prior work on the development of timely diagnosis tools has shown significant benefits in applying structured approaches to population-based screening efforts [74], [75]. Such diagnosis of asymptomatic patients is critical as it enables earlier intervention, which has been consistently shown to improve treatment outcomes for various diseases [22], [76]. Machine learning approaches have further enhanced the effectiveness of this screening in specific diseases, including breast and lung cancer [25], [77].

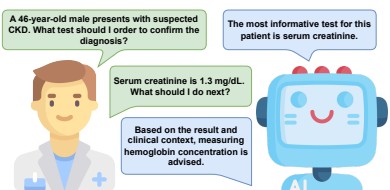

Figure 8: Example of a ML assistant guiding a clinician through sequential diagnostic refinement for a suspected chronic kidney disease (CKD) case. The agent recommends the most informative next test and iteratively updates its suggestions as new results arrive.

Despite these valuable applications and advances, a unified Bayesian framework that fully supports the entire sequential decision-making process inherent in clinical diagnosis, from the initial formation of hypotheses to the targeted acquisition and sophisticated integration of heterogeneous clinical data, remains largely underdeveloped. Our current work directly addresses this critical gap by demonstrating how large language models (LLMs) can effectively serve as powerful surrogates for Bayesian Experimental Design, thereby providing a novel

computational approach to support and improve real-world diagnostic tasks by facilitating this comprehensive sequential reasoning and data integration process.

### Active learning in medicine

Active learning is a subfield of machine learning in which models strategically select the most informative data points to observe, thereby minimizing the need for costly human labelling [78]. It has been successfully applied in both unsupervised settings to identify informative features [79] and in conjunction with deep learning architectures to improve data efficiency [80]. Central to active learning is the definition of acquisition functions or utility metrics that quantify the expected benefit of observing new data. Large language models (LLMs) have recently been explored as surrogate models for guiding such selection, particularly in settings where data acquisition is expensive or sparse [81]. In biomedicine, active learning has been applied to optimize test ordering and experimental design, where the cost of procedures is often a limiting factor. For example, in pharmacology, it has been used to prioritize compound testing [82], and in clinical treatment planning, active feature selection methods have enabled more efficient personalization of care [83]. Despite these advances, the use of LLMs for active test selection in the context of sequential clinical diagnosis remains unexplored. Figure 8 illustrates how such an ML-based framework can support clinicians by optimizing the selection of diagnostic tests to sequentially refine the diagnosis.

## B   Extended works

### Summary of formalism

A summary of the mathematical formalism introduced is given in Table 5.

Table 5: Summary of notation used in the agent-based diagnosis and Bayesian experimental design framework.

| Symbol | Definition |
| --- | --- |
| $\Sigma$ | Space of natural language strings |
| $\mathcal{S} \in \Sigma$ | Natural language input or instruction to the agent |
| $\mathcal{D}_t \subset \Sigma$ | Set of all possible diagnoses at time $t$ |
| $\mathcal{D}_{\text{true},t} \subset \mathcal{D}_t$ | Subset of true diagnoses for a patient at time $t$ |
| $d_t \in \mathcal{D}_t$ | Single diagnosis considered at time $t$ |
| $T = \{1, 2, \ldots, T_{\max}\}$ | Discrete time horizon for the decision process |
| $\mathcal{X}_t \subset \Sigma$ | Ground-truth information available at time $t$ |
| $K_t \subset \mathcal{X}_t$ | Known (observed) information at time $t$ |
| $\mathcal{U}_t = \mathcal{X}_t \setminus K_t$ | Unknown (unobserved) information at time $t$ |
| $U_t^{(i)}$ | Random variable representing candidate diagnostic test $i$ at time $t$ |
| $u_t^{(i,j)} \sim P(U_t^{(i)})$ | Sampled realization $j$ from candidate test $i$ |
| $\hat{u}_t^{(i)}$ | Observed (true) outcome of test $i$ at time $t$ |
| $P(y_{d_t} = 1 \mid K_t)$ | Posterior belief over diagnosis $d_t$ given current knowledge $K_t$ |
| $\mathbb{H}[P(\cdot)]$ | Shannon entropy of a probability distribution |
| $\text{EIG}(U_t^{(i)})$ | Expected information gain of test $i$ |
| $p_{\text{prior}}$ | Prior belief distribution before test observation |
| $p_{\text{post}}$ | Posterior belief distribution after observing test result |
| $\text{KL}(q \parallel p)$ | Kullback–Leibler divergence between two distributions $q$ and $p$ |
| $M$ | Number of Monte Carlo samples used to estimate expectations |
| $\mathcal{F}(U_t^{(i)})$ | Utility or information-theoretic value of diagnostic test $i$ |
| $c(U_t^{(i)})$ | Cost associated with performing test $U_t^{(i)}$ |
| $\lambda$ | Trade-off parameter weighting diagnostic accuracy vs. test cost |
| $\mathcal{L}(y, u_t, \lambda)$ | Lagrangian objective combining diagnostic error and cumulative test cost |

**Formulation of the Bayesian Experimental Design (BED) Objective using KL Divergence**

**Epistemic vs. Aleatoric Uncertainty in Active Learning.** In standard Active Learning (AL), which represents a common instance of BED, the goal is to reduce *epistemic* uncertainty by querying the label of the most informative observation $x_o$. The expected information gain can be written as

$$\underbrace{I(Y; Z \mid x_o, d)}_{\text{epistemic}} = H(Y \mid d, x_o) - \underbrace{H(Y \mid Z, d, x_o)}_{\text{aleatoric}}, \tag{3}$$

where $Z$ is a random variable representing the latent hypothesis or model parameters, $d$ denotes the currently observed data, $H(\cdot)$ is Shannon entropy, and $I(\cdot)$ denotes mutual information.

The first term quantifies the total predictive uncertainty, while the second term captures the irreducible (aleatoric) uncertainty given the true latent model. Their difference isolates the epistemic component:

$$\text{Epistemic} = \text{Predictive} - \text{Aleatoric}.$$

**Active Feature Acquisition (AFA).** Our clinical test selection setting is more closely related to *Active Feature Acquisition* (AFA), where the agent selects the next feature or diagnostic test $F_j$ to maximally reduce uncertainty about the disease label $Y$. The corresponding mutual information criterion is

$$I(Y; F_j) = H(Y) - H(Y \mid F_j). \tag{4}$$

Since $H(Y)$ is constant across candidate features, maximizing $I(Y; F_j)$ is equivalent to minimizing $H(Y \mid F_j)$; thus, the most informative test is the one that most reduces predictive entropy.

**KL-Based Expected Information Gain.** Information gain can be equivalently expressed in entropy or KL-divergence space. Equation (2) in the main text evaluates the expected divergence between posterior and prior predictive distributions as

$$\mathbb{E}[\text{KL}] = -H(Y \mid F_j) + \text{CE}(Y, Y_{\text{prior}}), \tag{5}$$

where CE denotes cross-entropy between the posterior predictive distribution and an uninformative prior.

**Interpretation.**

- **Term A** ($-H(Y \mid F_j)$): corresponds to the entropy-reduction term in AFA, rewarding features that reduce predictive uncertainty.
- **Term B** ($\text{CE}(Y, Y_{\text{prior}})$): encourages divergence from the prior, preventing redundant queries by favoring features that meaningfully update prior beliefs.

**Note on notation.** The prior probability $p_{\text{prior}}$ is constant and no longer indexed by $j$, a correction applied in the current manuscript revision.

**Concrete Derivation** For a single Monte Carlo sample, the KL divergence between the posterior and prior disease probabilities is computed as

$$\text{KL}(p_{\text{post}} \| p_{\text{prior}}) = p_{\text{post}} \log \frac{p_{\text{post}}}{p_{\text{prior}}} + (1 - p_{\text{post}}) \log \frac{1 - p_{\text{post}}}{1 - p_{\text{prior}}}. \tag{6}$$

Rearranging terms yields

$$\text{KL}(p_{\text{post}} \| p_{\text{prior}}) = \underbrace{\left[ p_{\text{post}} \log p_{\text{post}} + (1 - p_{\text{post}}) \log(1 - p_{\text{post}}) \right]}_{\text{Term A: } -H(Y \mid F_j)}$$
$$- \underbrace{\left[ p_{\text{post}} \log p_{\text{prior}} + (1 - p_{\text{post}}) \log(1 - p_{\text{prior}}) \right]}_{\text{Term B: } \text{CE}(Y, Y_{\text{prior}})}. \tag{7}$$

Averaging over $M$ Monte Carlo samples gives the final expected value used in Eq. (2):

$$\mathbb{E}_{m=1}^{M} \left[ \text{KL}(p_{\text{post}}^{(m)} \| p_{\text{prior}}) \right]. \tag{8}$$

This formulation provides a principled Bayesian information-gain objective that decomposes into uncertainty reduction and prior divergence terms, bridging the entropy-based and KL-based views of active test selection.

**Directly calculating the EIG from entropy**

The information gain (IG) from an additional piece of information $u_t^{(i,j)}$ is defined as:

$$\text{IG}(u_t^{(i)}) := \mathbb{H}[P(y_d = 1 \mid K_t)] - \mathbb{H}[P(y_d = 1 \mid K_t, u_t^{(i,j)})]. \tag{9}$$

where $\mathbb{H}$ denotes Shannon entropy. The first term represents the uncertainty about the diagnosis $D_t$ given the current knowledge $K_t$, while the second term represents the uncertainty after observing the additional piece of information $u_t^{(i)}$. Since this quantity is difficult to compute directly, we approximate it using an expectation over samples to obtain the Expected Information Gain (EIG):

$$\text{EIG}(u_t^{(i)}) = \mathbb{H}[P(y_d = 1 \mid K_t)] - \mathbb{E}_{u_t^{(i,j)} \sim P(u_t^i)} \left[ \mathbb{H}[P(y_d = 1 \mid K_t, u_t^{(i,j)})] \right]. \tag{10}$$

While it is possible to directly estimate the expected information gain (EIG) using changes in entropy, this approach can sometimes favour diagnostic tests that are not truly informative. For binary classification tasks, we model the posterior distribution $P(y_d = 1 \mid K_t, u_t^{(i)})$ as a Bernoulli distribution $\mathbb{B}(p_i)$ with success probability $p_i \in [0, 1]$. The entropy of a Bernoulli distribution is given by:

$$\mathbb{H}[\mathbb{B}(p_i)] = -p_i \log(p_i) - (1 - p_i) \log(1 - p_i). \tag{11}$$

We approximate the expected entropy using samples $u_t^{(i,j)} \sim P(\mathcal{U}_t)$ and the corresponding posterior probabilities $p_{i,j} = P(y_d = 1 \mid K_t, u_t^{(i,j)})$:

$$\text{EIG}(u_t^{(i)}) \approx \mathbb{H}[P(y_d = 1 \mid K_t)] - \frac{1}{M} \sum_{j=1}^{M} \left( -p_{i,j} \log p_{i,j} - (1 - p_{i,j}) \log(1 - p_{i,j}) \right). \tag{12}$$

Finally, the optimal piece of information to query is:

$$i^* = \arg \max_i \text{EIG}(u_t^{(i)}). \tag{13}$$

However, entropy-based selection can inadvertently prefer tests that preserve an already confident belief—even when that belief is wrong—over tests that challenge it. Consider a patient with an initial disease probability of 0.1. A truly informative test might raise this probability to 0.6, while an uninformative test would leave it at 0.1 regardless of outcome. Because entropy is low near the extremes, an entropy-based EIG can assign little value to the informative test (which moves the belief away from a low-entropy region) and may instead tolerate the non-informative one that keeps the posterior near 0.1. In contrast, an

Table 6: *KL vs. entropy selection.* Mean accuracy (± std) across datasets.

| Model | Dataset | KL-based | Entropy-based |
|---|---|---|---|
| GPT-4o-mini | Diabetes | $0.593 \pm 0.039$ | $0.572 \pm 0.005$ |
| GPT-4o-mini | Kidney | $0.992 \pm 0.006$ | $0.972 \pm 0.010$ |
| GPT-4o-mini | Hepatitis | $0.825 \pm 0.013$ | $0.770 \pm 0.010$ |
| GPT-4o | Diabetes | $0.682 \pm 0.012$ | $0.673 \pm 0.011$ |
| GPT-4o | Kidney | $0.999 \pm 0.003$ | $0.994 \pm 0.000$ |
| GPT-4o | Hepatitis | $0.839 \pm 0.013$ | $0.770 \pm 0.017$ |

expected Kullback–Leibler (KL) divergence criterion explicitly measures the magnitude of change in the predictive distribution and thus correctly prioritizes the test that meaningfully shifts beliefs. Empirically, our KL-based selection outperforms the entropy-based variant across datasets (Table 6).

## C   Experimental details

**Datasets**

We evaluated the performance of our model on three clinical datasets of varying complexity:

**Chronic Kidney Disease.**   The first task involves the prediction of chronic kidney disease (CKD) from symptoms and laboratory results [84]. The model is provided with demographic variables such as age and gender, as well as signs observable on physical examination (e.g., oedema). It is then tasked with diagnosing CKD by selectively ordering lab tests such as serum albumin or serum creatinine. We filtered the dataset to retain only instances without missing values, resulting in a total of 157 patients. Categorical lab tests lacking a clear clinical interpretation, such as Pus Cells: Abnormal, were excluded. The dataset is available from the UCI Machine Learning Repository under a CC BY 4.0 license: https://archive.ics.uci.edu/dataset/336/chronic+kidney+disease.

**Hepatitis.** The second task requires the model to diagnose Hepatitis C virus (HCV) infection using liver function test results [85]. Age and sex were provided as known demographic features, while laboratory tests (e.g., ALT, AST, GGT) could be queried by the model. We included all 56 patients with confirmed HCV and selected a random subset of 56 healthy individuals to form a balanced dataset. The dataset is publicly available from the UCI Repository under a CC BY 4.0 license: https://archive.ics.uci.edu/dataset/571/hcv+data.

**Diabetes.** The third task uses a random subset of 100 patients from the Pima Indians Diabetes dataset, originally collected by the National Institute of Diabetes and Digestive and Kidney Diseases [86]. The model is asked to predict the presence of diabetes based on clinical measurements. All individuals are female, at least 21 years old, and of Pima Indian heritage. Age is treated as a known feature; the remaining features are unknown and can be selectively queried. The dataset is available on Kaggle under a CC0 Public Domain license: https://www.kaggle.com/datasets/uciml/pima-indians-diabetes-database.

**OSCE.** We construct a custom OSCE-style dataset derived from *AgentClinic* [54] to evaluate zero-shot generalization in a multi-condition setting. The cohort comprises 114 clinically diverse cases spanning a range of presentations and diseases. To align with our focus on sequential test acquisition, we retain only laboratory-style diagnostic features and the ground-truth diagnosis. The model is tasked with predicting the primary diagnosis given limited initial context (demographics and brief presentation), while remaining lab tests are unknown and can be selectively queried. For each case, we additionally create a matched synthetic counterpart by replacing disease-indicative labs with clinically normal values that point away from the true diagnosis, yielding case–control pairs for evaluating test selection policies. We release the modified OSCE dataset alongside our code to facilitate replication and comparison.

### Input data formatting

LLMs have been shown to struggle interpreting tabular clinical data accurately [38]. To mitigate this limitation, we converted all available clinical information from structured tabular form into concise natural language vignettes. Each vignette integrates demographic information, diagnostic test results, and measurement units where appropriate, providing a more interpretable and context-rich format for the model. Table 7 presents an example of the raw tabular input used for diagnosing chronic kidney disease. This input is automatically converted into a clinical vignette using a rule-based preprocessing script (see Clinical Vignette 1).

---

**Clinical Vignette 1**

The patient is 63 years old. The patient's diastolic blood pressure is 70 mm/Hg. The patient has a poor appetite. The patient has pedal oedema. The patient has hypertension. The patient has diabetes mellitus. The patient does not have coronary artery disease. The patient does not have anaemia. Specific gravity was measured at 1.01. Albumin levels in urine was measured at 3.0. Sugar levels in urine was measured at 0.0. Blood glucose random was measured at 380.0 mg/dL. Blood urea was measured at 60.0 mg/dL. Serum creatinine was measured at 2.7 mg/dL. Sodium levels was measured at 131.0 mEq/L. Potassium levels was measured at 4.2 mEq/L. Haemoglobin levels was measured at 10.8 g/dL. Packed cell volume was measured at 32.0.

---

### Experiment

For each patient in a preprocessed dataset subset, we evaluated risk predictions using all features, as well as under four feature selection strategies: Bayesian, Random, Global Best (predefined), and Implicit. At each iteration, one additional feature from the unknown set was revealed, and corresponding risks were computed. Pseudocode for the Bayesian selection using the KL-divergence is given in Algorithm 1. Importantly, the risk prediction prompt remained unchanged between feature selection methods other than the clinical information added at the end. All experiments were implemented using GPT-4o (Version 2024-11-20) and GPT-4o-mini (Version 2024-07-18) as provided on the Azure OpenAI Service. To ensure robustness, each experiment was run across 5 different random seeds. Opens-source experiments were run suing Biomistral-7B and Llama70B version 3.3 as provided on HuggingFace using vLLM. For all experiments, we set the number of sampled test outcomes or risk probability distributions to 10. To ensure the sampling produced a more diverse set of responses, we used a temperature of 1 and specifically instructed the model in the prompts to simulate randomness. Performance metrics were averaged across runs with their corresponding standard deviation.

## D  LLM Prompts

For each dataset, we use four distinct prompt types. The first prompt, shared across all methods, is used for risk prediction based on known patient data, ensuring consistency in evaluation. The second prompt is issued at the start of the experiment, before observing any patient-specific information, to identify globally optimal features.

Table 7: Structured clinical data for an example patient prior to natural language transformation and inclusion of appropriate units.

| Category | Test Name | Result |
|---|---|---|
| **Demographics** | Age | 63 |
| **Vital Signs** | Blood Pressure | 70 |
| **Urine Tests** | Specific Gravity | 1.010 |
| | Albumin | 3 |
| | Sugar | 0 |
| | Red Blood Cells | Abnormal |
| | Pus Cells | Abnormal |
| | Clumps | Present |
| | Bacteria | Not Present |
| **Blood Tests** | Blood Glucose | 380 |
| | Blood Urea | 60 |
| | Creatinine | 2.7 |
| | Haemoglobin | 10.8 |
| | PCV | 32 |
| | WBC Count | 4500 |
| | RBC Count | 3.8 |
| **Electrolytes** | Sodium | 131 |
| | Potassium | 4.2 |
| **Symptoms** | Appetite | Poor |
| | Pedal Edema | Yes |
| | Anemia | No |
| **Comorbidities** | Hypertension | Yes |
| | Diabetes Mellitus | Yes |
| | Coronary Artery Disease | No |

---

**Algorithm 1** KL-guided Diagnostic Test Selection

---

**Require:** Initial knowledge $K_t$, unknowns $\{u_t^{(i)}\}_{i=1}^N$, number of samples $M$

1: Sample $M$ draws from $P(y_{d_t} = 1 \mid K_t)$: $\{p_{\text{prior}}^{(j)}\}_{j=1}^M$
2: **for** $i = 1$ to $N$ **do**
3:     **for** $j = 1$ to $M$ **do**
4:         Sample $u_t^{(i,j)} \sim P(u_t^{(i)})$
5:         Compute posterior: $p_{\text{posterior}}^{(j)} = P(y_{d_t} = 1 \mid K_t, u_t^{(i,j)})$
6:         Compute $\text{KL}^{(j)} = p_{\text{prior}}^{(j)} \log \frac{p_{\text{prior}}^{(j)}}{p_{\text{posterior}}^{(j)}} + (1 - p_{\text{prior}}^{(j)}) \log \frac{1 - p_{\text{prior}}^{(j)}}{1 - p_{\text{posterior}}^{(j)}}$
7:     **end for**
8:     $\mathbb{E}[\text{KL}(u_t^{(i)})] = \frac{1}{M} \sum_{j=1}^M \text{KL}^{(j)}$
9: **end for**
10: $i^* = \arg\max_i \mathbb{E}[\text{KL}(u_t^{(i)})]$
11: Request $\hat{u}_t^{(i^*)}$ and update $K_{t+1} = K_t \cup \{\hat{u}_t^{(i^*)}\}$

---

The third prompt is used in the implicit selection baseline, where the model is queried at each decision step to choose the most suitable test based on the information observed so far. The final prompt is specific to the Bayesian Experimental Design setting and is used to sample a plausible outcome for a candidate diagnostic test, conditioned on the currently available patient data.

---

**Risk Prediction Prompt: CKD**

You are an expert nephrologist. Based on the following clinical data and the patient's history, provide an estimate of the patient having chronic kidney disease as a single number between 0 and 1. Consider key laboratory markers and other pertinent values. When these values indicate kidney disease or damage—assign a number closer to 1, indicating a higher probability of chronic kidney disease. If the laboratory values are within normal ranges, assign a value closer to 0. Return only the number that can be converted to a Python `float`, without any additional commentary.
`$known_info`

---

**Global Feature Selection Prompt: CKD**

You are an expert nephrologist. Based on the following list of features: `$all_features`, please indicate which `$n` features you believe are the most informative and critical for diagnosing chronic kidney disease. Return your answer as a Python list of exactly `$n` feature names.
For example: if n is 1, return `['age']`; if n is 2, return `['age', 'bp']`.
Do not include any additional commentary.

---

**Feature Selection Prompt: CKD (Implicit)**

You are an expert nephrologist. Based solely on the following known clinical data, determine which additional feature from the list below would be the most informative to sample next for diagnosing chronic kidney disease.
**Known Data:** `$known_data`
**Unknown Features:** `$unknown_features`
Return only the name of the feature strictly in the form shown in the list as a string, without any additional commentary.

---

**Sampling Prompt: CKD Lab Test Simulation**

You are an expert nephrologist. Based on the following clinical data and the patient's history, please simulate a random draw from the full range of clinically plausible values for `$feature_to_sample`.
The value should not simply be the average or a central tendency, but should vary as if sampled at random from a realistic distribution. Consider the following description: `$ref_info`.
Avoid returning the same value repeatedly across multiple draws, and ensure the value varies as if sampled from a plausible distribution. Introduce randomness by considering edge cases, typical values, and outliers within the plausible range.
Return your answer as a single numeric value that can be parsed into a `float` in Python, with no additional commentary or units.
**IMPORTANT:** Assume that the patient may or may not have chronic kidney disease, and your sampling should reflect that uncertainty.
`$known_info`
**IMPORTANT:** Under NO circumstances provide explanations, commentary, or text beyond the single numeric float or string requested. The response MUST be parseable strictly as a float, e.g., 0.512, with no extra words. If a string is requested no float is required.

---

**Risk Prediction Prompt: Hepatitis C**

You are an expert hepatologist. Based on the following clinical data and the patient's history, please provide an estimate of the patient's risk of being infected with hepatitis C as a single number between 0 and 1. Consider key laboratory markers and other pertinent values. When these values indicate liver inflammation or damage — assign a number closer to 1, indicating a higher probability of hepatitis C infection. If the laboratory values are within normal ranges, assign a value closer to 0. Return only the number that can be converted to a Python float, without any additional commentary.
`$known_info`

**Global Feature Importance Prompt: Hepatitis C**

You are an expert hepatologist. Based on the following list of features: `$all_features`, please indicate which `$n` features you believe are the most informative and critical for diagnosing hepatitis.
Return your answer as a Python list of exactly `$n` feature names (for example, if n is 1, return `['ALT']`; if n is 2, return `['ALT', 'AST']`), without any additional commentary.
`$known_info`

---

**Feature Selection Prompt: Hepatitis C (Implicit)**

You are an expert hepatologist. Based solely on the following known clinical data, determine which additional feature from the list below would be the most informative to sample next for diagnosing hepatitis.
**Known Data:** `$known_data`
**Unknown Features:** `$unknown_features`
Return only the name of the feature as a string, without any additional commentary.

---

**Sampling Prompt: Hepatitis C Lab Test Simulation**

You are an expert hepatologist. Based on the following clinical data and the patient's history, please simulate a random draw from the full range of clinically plausible values for `$feature_to_sample`.
Consider the possible range as described: `$ref_info`. Ensure that the value you return is realistic and reflects clinical variability. Avoid returning the same value repeatedly across multiple draws, and ensure the value varies as if sampled from a plausible distribution. Introduce randomness by considering edge cases, typical values, and outliers within the plausible range.
Return your answer as a single numeric value that can be converted to a Python float, without any additional commentary.
**IMPORTANT:** Assume that the patient may or may not have hepatitis C, and your sampling should reflect that uncertainty.
`$known_info`
**IMPORTANT:** Under NO circumstances provide explanations, commentary, or text beyond the single numeric float or string requested. The response MUST be parseable strictly as a float, e.g., 0.512, with no extra words. If a string is requested no float is required.

---

**Risk Prediction Prompt: Diabetes**

You are an expert endocrinologist. Based on the following clinical data and the patient's history, provide an estimate of the patient's risk of diabetes as a single number between 0 and 1.
It is known that all patients are females at least 21 years old of Pima Indian heritage. Focus on key markers. Assign a value closer to 1 if the data indicate high risk, and closer to 0 if within normal limits. Return only the number, without any additional commentary.
`$known_info`

---

**Global Feature Importance Prompt: Diabetes**

You are an expert endocrinologist. Based on the following list of features: `$all_features`, please indicate which `$n` features you believe are the most informative and critical for diagnosing diabetes.
Return your answer as a Python list of exactly `$n` feature names (for example, if n is 1, return `['Glucose']`; if n is 2, return `['Glucose', 'BMI']`), without any additional commentary.

---

**Feature Selection Prompt: Diabetes (Implicit)**

You are an expert endocrinologist. Based solely on the following known clinical data, determine which additional feature from the list below would be the most informative to sample next for diagnosing diabetes.
**Known Data:** `$known_data`
**Unknown Features:** `$unknown_features`
Return only the name of the feature as a string, without any additional commentary.
`$known_info`

**Sampling Prompt: Diabetes Lab Test Simulation**

You are an expert endocrinologist. Based on the following clinical data and the patient's history, please simulate a random draw from the full range of clinically plausible values for `$feature_to_sample`.
Consider the following unit for the sampled value: `$ref_info`. Ensure that the value you return is realistic and reflects clinical variability. Avoid returning the same value repeatedly across multiple draws, and ensure the value varies as if sampled from a plausible distribution. Introduce randomness by considering edge cases, typical values, and outliers within the plausible range.
Return your answer as a single numeric value that can be converted to a Python float with no units or additional commentary.
**IMPORTANT:** Assume that the patient may or may not have diabetes, and your sampling should reflect that uncertainty.
`$known_info`
**IMPORTANT:** Under NO circumstances provide explanations, commentary, or text beyond the single numeric float or string requested. The response MUST be parseable strictly as a float, e.g., 0.512, with no extra words. If a string is requested no float is required.

---

**Risk Prediction Prompt: OSCE**

You are an expert clinician. Given the following case details, estimate a realistic and conservative probability of the suspected diagnosis of `$potential_diagnosis` as a single number between 0 and 1. Return only the number that can be converted to a Python float, without any additional commentary.
`$known_info`

---

**Global Feature Importance Prompt: OSCE**

You are a clinical assistant. Given these case details: `$known_data`. Which additional feature from the list `$unknown_features` would be most informative next for the suspected diagnosis of `$potential_diagnosis`? Return only the feature name without commentary.

---

**Feature Selection Prompt: OSCE (Implicit)**

You are a clinical assistant. Given these case details: `$known_data`, which feature from the list `$unknown_features` would provide the most information next? Return only the feature name without commentary."

---

**Sampling Prompt: OSCE Lab Test Simulation**

You are a expert clinician. Based on the following case details, simulate a plausible value for `$feature_to_sample` Return only the value without explanation."
**IMPORTANT:** Assume that the patient may or may not have diabetes, and your sampling should reflect that uncertainty.
`$known_info`
**IMPORTANT:** Under NO circumstances provide explanations, commentary, or text beyond the single numeric float or string requested. The response MUST be parseable strictly as a float, e.g., 0.512, with no extra words. If a string is requested no float is required.

# E Extended results

## Predicting plausible test outcome distributions

To evaluate model performance, we analyzed how the mean absolute error (MAE) changed with increasing numbers of generated samples. For computational efficiency, we limited the number of samples to 10 per query. As shown in Figure 9, we observe a consistent decrease in MAE as more features are acquired, indicating that the model is producing a diverse set of samples, some of which closely approximate the ground truth. Furthermore, the larger GPT-4o model outperformed the smaller GPT-4o-mini variant across all three datasets, highlighting the benefits of scale in both predictive accuracy and sample quality.

To assess the ability of large language models (LLMs) to generate diverse samples from hypothetical test outcome distributions, we analysed the generated values for all numerical features across the three datasets. Categorical features were excluded from this analysis. For numerical features, the LLM produced non-deterministic outputs that varied meaningfully between samples, rather than issuing static or averaged responses. This behaviour

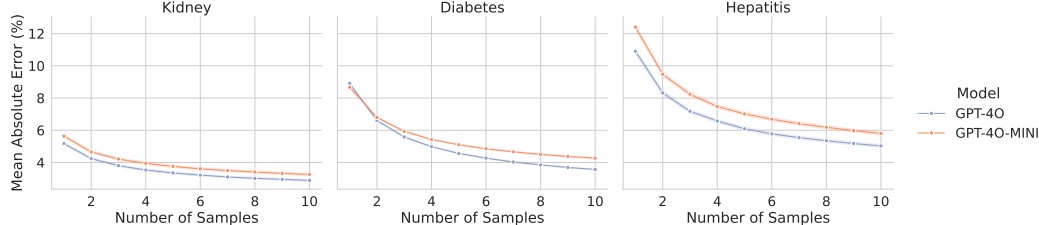

Figure 9: *Increasing sample size improves the MAE.* Model performances are averaged across 5 different seeds and are shown with their corresponding standard deviations.

indicates that the model is capable of representing distributional uncertainty in a clinically meaningful way. This behaviour was consistent across all three datasets, as illustrated in Figures 10, 11 and 12.

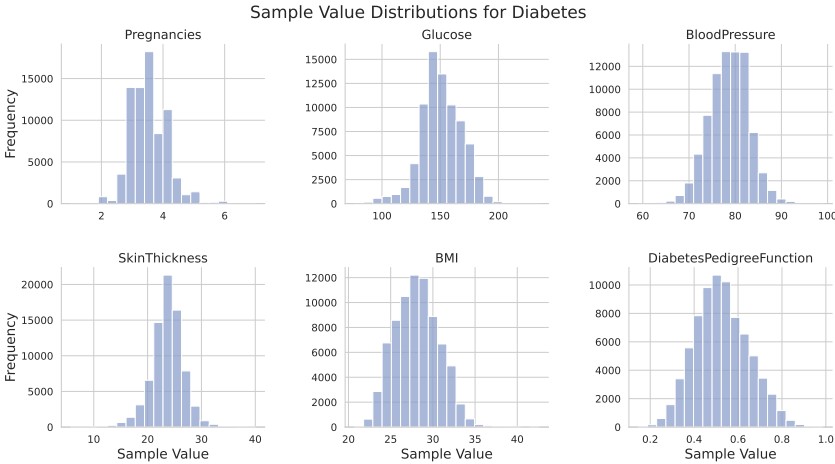

Figure 10: *Sample variability of LLM predictions in the diabetes dataset.* Each bar chart displays the distribution of values sampled for numerical features. The heterogeneity across samples indicates the model's capacity to avoid mode collapse and to reflect uncertainty consistent with clinical variation.

## Diagnostic Performance Evaluation

We evaluate `ACTMED` against multiple baselines using accuracy, precision, recall, F1 score, and ROC-AUC (see Tables 8 and 9). Performance improves across all metrics when moving from GPT-4o-mini to the more capable GPT-4o. On the kidney dataset, differences between models are minimal, suggesting both models perform reliably. In contrast, on the hepatitis and diabetes datasets, `ACTMED` consistently outperforms baseline feature selection methods and the full-feature classifier, demonstrating its ability to adaptively select informative subsets.

We further assess `ACTMED` under varying feature budget constraints (Table 10), controlled via the stopping threshold $\gamma$. Across all datasets and both models, diagnostic accuracy remains stable even as $\gamma$ varies. However, stricter thresholds (higher $\gamma$) lead to more conservative test acquisition and substantially fewer tests. For example, on the kidney dataset, both models typically require only 1–2 tests per patient when using the stopping criterion. At a conservative threshold of $\gamma = 0.7$, GPT-4o achieves near-perfect accuracy while querying just one test in nearly all cases.

## Feature selection impact on performance

This section investigates the impact of feature selection frameworks on diagnostic performance by analysing the features most frequently selected and their empirical informativeness. Using a random baseline to mitigate confounding, we first identified empirically informative features based on their accuracy when selected among three features. For GPT-4o-mini, the most informative features were GGT, AST, and ALT for hepatitis; BloodPressure, Pregnancies, and Insulin for diabetes; and Random Blood Glucose, Potassium Levels, and Serum Creatinine for CKD. For GPT-4o, BIL, CHE, and GGT were most informative for hepatitis; BloodPressure, Skin Thickness, and Insulin for diabetes; and Blood Glucose, Red Blood Cell Count, and Potassium Levels for CKD.

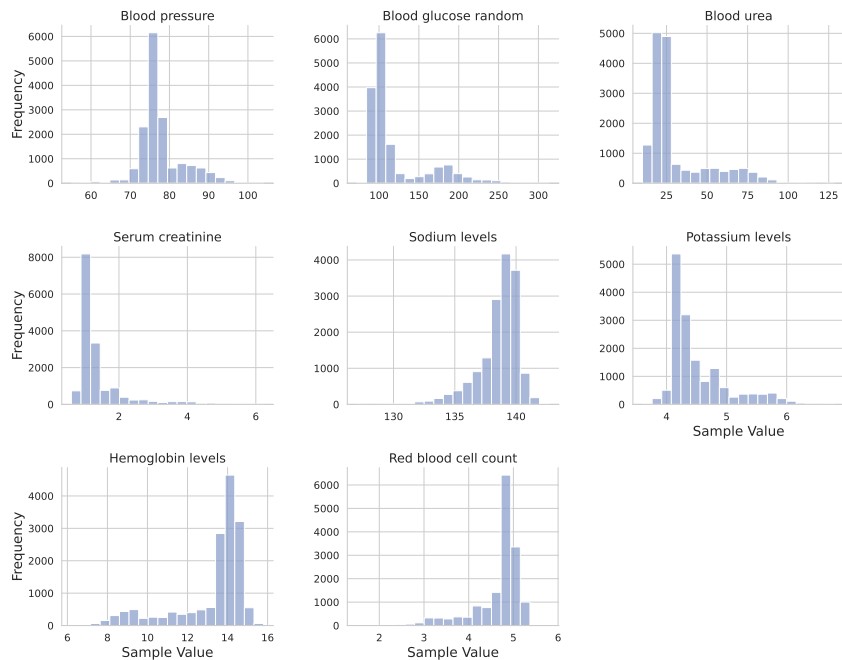

Figure 11: *Sampled feature distributions in the chronic kidney disease (CKD) dataset.* The LLM produces diverse and physiologically grounded value ranges across all features. This supports its utility as a surrogate model in capturing uncertainty for Bayesian evidence diagnostics.

Table 8: *Performance metrics (mean ± std) for GPT-4o-mini across datasets.* Best-performing methods are in **bold**; if All Features is best, the second-best is also indicated.

| Dataset | Method | AUC | Accuracy | F1 | Precision | Recall |
|---|---|---|---|---|---|---|
| Kidney | All Features | $0.9997 \pm 0.0002$ | $0.975 \pm 0.006$ | $0.956 \pm 0.009$ | $0.915 \pm 0.017$ | $\mathbf{1.000 \pm 0.000}$ |
| | **ACTMED (ours)** | $0.9999 \pm 0.0001$ | $0.992 \pm 0.006$ | $0.986 \pm 0.011$ | $0.978 \pm 0.020$ | $0.995 \pm 0.009$ |
| | Random | $0.9990 \pm 0.0002$ | $0.981 \pm 0.004$ | $0.966 \pm 0.007$ | $0.947 \pm 0.021$ | $0.986 \pm 0.011$ |
| | Global Best | $\mathbf{1.000 \pm 0.000}$ | $\mathbf{1.000 \pm 0.000}$ | $\mathbf{1.000 \pm 0.000}$ | $\mathbf{1.000 \pm 0.000}$ | $\mathbf{1.000 \pm 0.000}$ |
| | Implicit | $\mathbf{1.000 \pm 0.000}$ | $0.997 \pm 0.005$ | $0.995 \pm 0.009$ | $0.991 \pm 0.018$ | $\mathbf{1.000 \pm 0.000}$ |
| Hepatitis | All Features | $0.874 \pm 0.002$ | $0.793 \pm 0.009$ | $0.758 \pm 0.011$ | $\mathbf{0.910 \pm 0.012}$ | $0.650 \pm 0.014$ |
| | **ACTMED (ours)** | $\mathbf{0.891 \pm 0.006}$ | $\mathbf{0.825 \pm 0.013}$ | $\mathbf{0.806 \pm 0.015}$ | $0.903 \pm 0.017$ | $\mathbf{0.729 \pm 0.017}$ |
| | Random | $0.743 \pm 0.018$ | $0.670 \pm 0.020$ | $0.558 \pm 0.031$ | $0.843 \pm 0.045$ | $0.418 \pm 0.029$ |
| | Global Best | $0.824 \pm 0.007$ | $0.718 \pm 0.018$ | $0.668 \pm 0.022$ | $0.812 \pm 0.031$ | $0.568 \pm 0.021$ |
| | Implicit | $0.856 \pm 0.016$ | $0.770 \pm 0.007$ | $0.743 \pm 0.005$ | $0.843 \pm 0.023$ | $0.664 \pm 0.013$ |
| Diabetes | All Features | $0.727 \pm 0.011$ | $0.417 \pm 0.006$ | $0.545 \pm 0.003$ | $0.374 \pm 0.003$ | $\mathbf{1.000 \pm 0.000}$ |
| | **ACTMED (ours)** | $\mathbf{0.743 \pm 0.009}$ | $\mathbf{0.569 \pm 0.004}$ | $\mathbf{0.601 \pm 0.002}$ | $\mathbf{0.444 \pm 0.002}$ | $0.930 \pm 0.005$ |
| | Random | $0.663 \pm 0.008$ | $0.493 \pm 0.008$ | $0.560 \pm 0.005$ | $0.402 \pm 0.004$ | $0.923 \pm 0.004$ |
| | Global Best | $0.725 \pm 0.006$ | $0.462 \pm 0.003$ | $0.563 \pm 0.001$ | $0.393 \pm 0.001$ | $0.992 \pm 0.001$ |
| | Implicit | $0.733 \pm 0.004$ | $0.460 \pm 0.002$ | $0.562 \pm 0.001$ | $0.392 \pm 0.001$ | $0.993 \pm 0.000$ |
| OSCE | All Features | $0.755 \pm 0.009$ | $0.684 \pm 0.015$ | $0.744 \pm 0.012$ | $0.625 \pm 0.010$ | $0.919 \pm 0.018$ |
| | **ACTMED (ours)** | $0.755 \pm 0.028$ | $\mathbf{0.705 \pm 0.016}$ | $\mathbf{0.760 \pm 0.014}$ | $\mathbf{0.641 \pm 0.011}$ | $\mathbf{0.933 \pm 0.023}$ |
| | Random | $0.653 \pm 0.019$ | $0.604 \pm 0.020$ | $0.670 \pm 0.013$ | $0.575 \pm 0.018$ | $0.804 \pm 0.026$ |
| | Global Best | $0.756 \pm 0.018$ | $0.667 \pm 0.021$ | $0.735 \pm 0.017$ | $0.610 \pm 0.014$ | $0.923 \pm 0.024$ |
| | Implicit | $0.727 \pm 0.026$ | $0.665 \pm 0.007$ | $0.732 \pm 0.006$ | $0.610 \pm 0.004$ | $0.916 \pm 0.013$ |

We also identified globally optimal features selected by both models as Glucose, BMI, and Insulin for diabetes, ALT, AST, and BIL for hepatitis, and BloodPressure, Serum Creatinine, and Haemoglobin for CKD.

We then examined the feature selection patterns of `ACTMED` and the implicit baseline. The random baseline served as a control, showing no significant selection bias. For hepatitis, `ACTMED` with GPT-4o-mini preferentially selected ALT, AST, and GGT, while with GPT-4o it favoured AST, GGT, and BIL. The implicit method selected ALT, AST, and GGT for GPT-4o-mini and ALT, AST, and BIL for GPT-4o. On the diabetes dataset, GPT-4o-

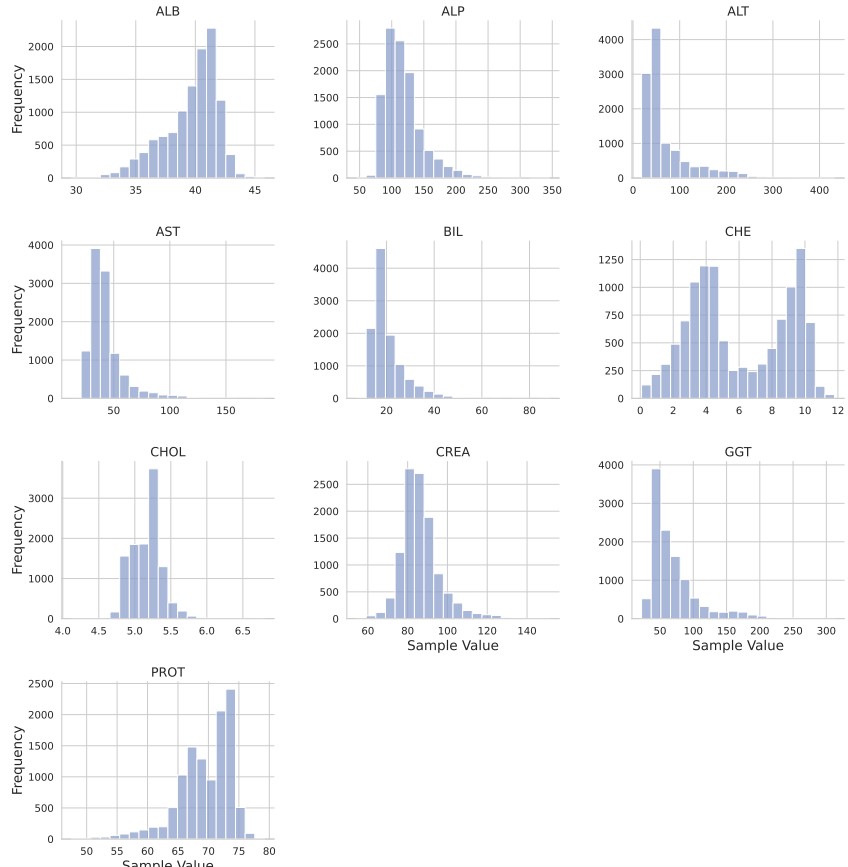

Figure 12: *Distributions of LLM-generated samples for numerical features in the hepatitis dataset.* Bar plots show the spread of sampled values for each feature, illustrating the model's ability to represent plausible clinical variability. The distributions vary across features, reflecting both physiological ranges and disease-specific uncertainty.

mini with `ACTMED` selected Glucose, Blood Pressure, and Skin Thickness, whereas GPT-4o preferred Glucose, Diabetes Pedigree Function, and BMI. The implicit method consistently selected BMI, Glucose, and Insulin across both models. For the CKD dataset, GPT-4o-mini with `ACTMED` selected Red Blood Cell Count, Blood Pressure, and Creatinine, while GPT-4o selected Blood Urea, Creatinine, and Blood Pressure. The implicit method selected Serum Creatinine, Blood Urea, and Blood Pressure for GPT-4o-mini, and Serum Creatinine, Blood Urea, and Haemoglobin for GPT-4o.

Collectively, these results indicate that neither `ACTMED` nor the implicit method consistently selects a fixed set of features that are empirically superior across all scenarios. The observed performance advantage of `ACTMED` likely stems not from identifying a universally optimal feature subset, but rather from its ability to perform more varied and potentially better-personalized test selections compared to the more constrained implicit or global methods, thereby optimizing the diagnostic process for individual cases.

**Importance of LLM quality for `ACTMED`**

`ACTMED` is inherently model-agnostic, as the framework only requires access to a simulator capable of generating test outcomes and corresponding risk distributions. Our contribution is a test-time computational improvement — selecting tests to maximize information gain — rather than a modification of the underlying language model. Consequently, the primary consideration is not whether the model is open-source or closed-source, but whether it is capable of generating sufficiently accurate and diverse outcome distributions. To assess the impact of model quality, we evaluated two smaller open-source models, **BioMistral-7B** and **LLaMA-70B**, as representative lower-bound baselines.

Table 9: *Performance metrics (mean ± std) for GPT-4o across datasets.* Best-performing methods are in **bold**; if All Features is best, the second-best is also indicated.

| Dataset | Method | AUC | Accuracy | F1 | Precision | Recall |
|---|---|---|---|---|---|---|
| Kidney | All Features | **1.000 ± 0.000** | **1.000 ± 0.000** | **1.000 ± 0.000** | **1.000 ± 0.000** | **1.000 ± 0.000** |
| | ACTMED (ours) | **1.000 ± 0.000** | 0.999 ± 0.003 | 0.998 ± 0.005 | **1.000 ± 0.000** | 0.995 ± 0.009 |
| | Random | 0.999 ± 0.001 | 0.995 ± 0.003 | 0.991 ± 0.005 | **1.000 ± 0.000** | 0.981 ± 0.009 |
| | Global Best | **1.000 ± 0.000** | 0.999 ± 0.003 | 0.998 ± 0.005 | **1.000 ± 0.000** | 0.995 ± 0.009 |
| | Implicit | **1.000 ± 0.000** | 0.999 ± 0.003 | 0.998 ± 0.005 | **1.000 ± 0.000** | 0.995 ± 0.009 |
| Hepatitis | All Features | 0.876 ± 0.014 | 0.807 ± 0.004 | 0.771 ± 0.006 | **0.948 ± 0.001** | 0.650 ± 0.009 |
| | ACTMED (ours) | **0.917 ± 0.013** | **0.839 ± 0.013** | **0.814 ± 0.016** | 0.966 ± 0.012 | **0.704 ± 0.021** |
| | Random | 0.728 ± 0.039 | 0.657 ± 0.018 | 0.507 ± 0.036 | 0.904 ± 0.052 | 0.354 ± 0.035 |
| | Global Best | 0.799 ± 0.013 | 0.757 ± 0.007 | 0.699 ± 0.009 | 0.919 ± 0.011 | 0.564 ± 0.009 |
| | Implicit | 0.805 ± 0.013 | 0.729 ± 0.016 | 0.646 ± 0.030 | 0.927 ± 0.008 | 0.496 ± 0.036 |
| Diabetes | All Features | 0.815 ± 0.004 | 0.592 ± 0.004 | 0.623 ± 0.003 | 0.460 ± 0.003 | 0.967 ± 0.005 |
| | ACTMED (ours) | **0.803 ± 0.009** | **0.668 ± 0.009** | **0.648 ± 0.008** | **0.514 ± 0.008** | **0.878 ± 0.012** |
| | Random | 0.703 ± 0.011 | 0.581 ± 0.010 | 0.593 ± 0.007 | 0.448 ± 0.007 | 0.875 ± 0.014 |
| | Global Best | 0.781 ± 0.004 | 0.548 ± 0.005 | 0.599 ± 0.003 | 0.434 ± 0.003 | 0.969 ± 0.005 |
| | Implicit | 0.783 ± 0.003 | 0.550 ± 0.006 | 0.601 ± 0.003 | 0.435 ± 0.003 | 0.971 ± 0.003 |
| OSCE | All Features | 0.805 ± 0.006 | 0.719 ± 0.008 | 0.772 ± 0.007 | 0.650 ± 0.006 | **0.951 ± 0.013** |
| | ACTMED (ours) | **0.791 ± 0.015** | **0.746 ± 0.006** | **0.785 ± 0.008** | 0.680 ± 0.006 | 0.930 ± 0.029 |
| | Random | 0.696 ± 0.024 | 0.640 ± 0.021 | 0.701 ± 0.021 | 0.600 ± 0.015 | 0.842 ± 0.035 |
| | Global Best | 0.838 ± 0.011 | 0.723 ± 0.012 | 0.771 ± 0.010 | 0.657 ± 0.010 | 0.933 ± 0.020 |
| | Implicit | 0.730 ± 0.026 | 0.681 ± 0.016 | 0.736 ± 0.013 | 0.627 ± 0.012 | 0.891 ± 0.017 |

Table 10: *Average number of tests selected and accuracy (mean ± standard deviation) under KL-based termination for varying γ values.* Higher $\gamma$ requires stronger evidence to continue testing. Results are reported across five random seeds.

| Model | γ | Hepatitis | | Diabetes | | Kidney | |
|---|---|---|---|---|---|---|---|
| | | Tests | Accuracy | Tests | Accuracy | Tests | Accuracy |
| GPT-4O-MINI | 0.3 | 2.36 ± 0.68 | 0.832 ± 0.374 | 2.45 ± 0.71 | 0.560 ± 0.496 | 1.60 ± 0.75 | 0.997 ± 0.050 |
| | 0.5 | 1.87 ± 0.77 | 0.841 ± 0.366 | 2.09 ± 0.75 | 0.554 ± 0.497 | 1.48 ± 0.68 | 0.999 ± 0.036 |
| | 0.7 | 1.48 ± 0.73 | 0.845 ± 0.363 | 1.90 ± 0.74 | 0.542 ± 0.498 | 1.28 ± 0.52 | 0.997 ± 0.050 |
| GPT-4O | 0.3 | 2.67 ± 0.60 | 0.839 ± 0.368 | 2.27 ± 0.84 | 0.674 ± 0.469 | 1.35 ± 0.64 | 0.999 ± 0.036 |
| | 0.5 | 2.41 ± 0.67 | 0.827 ± 0.379 | 1.90 ± 0.89 | 0.683 ± 0.465 | 1.09 ± 0.35 | 0.996 ± 0.062 |
| | 0.7 | 2.27 ± 0.65 | 0.816 ± 0.388 | 1.69 ± 0.84 | 0.684 ± 0.465 | 1.04 ± 0.24 | 0.996 ± 0.062 |

Table 11: *Accuracy versus number of sequentially added tests.* Values denote mean ± standard deviation of predictive accuracy across evaluation folds.

| Dataset | Model | 1 Test | 3 Tests | 5 Tests | All Tests |
|---|---|---|---|---|---|
| Kidney | GPT-4o | 0.994 ± 0.004 | 0.995 ± 0.003 | 0.995 ± 0.003 | **1.000 ± 0.000** |
| | GPT-4o-mini | 0.975 ± 0.011 | 0.983 ± 0.005 | 0.990 ± 0.010 | 0.972 ± 0.006 |
| Hepatitis | GPT-4o | 0.589 ± 0.023 | 0.679 ± 0.022 | 0.718 ± 0.013 | **0.796 ± 0.010** |
| | GPT-4o-mini | 0.588 ± 0.030 | 0.698 ± 0.022 | 0.732 ± 0.010 | **0.798 ± 0.011** |
| Diabetes | GPT-4o | 0.730 ± 0.138 | 0.662 ± 0.170 | 0.664 ± 0.170 | 0.678 ± 0.161 |
| | GPT-4o-mini | 0.546 ± 0.015 | 0.524 ± 0.025 | 0.528 ± 0.016 | 0.458 ± 0.007 |

**Failure to generate accurate samples.** The **BioMistral-7B** model frequently fails to generate realistic outcome samples, often producing implausible outcomes such as non-integer values for the number of previous pregnancies. As a result, the observed Wasserstein distances between the model-generated and empirical distributions are substantially higher compared to **GPT-4o** and **GPT-4o-mini** (see Table 13).

**Failure to perform BED accurately.** The **LLaMA-70B** model also fails to achieve accuracy comparable to the closed-source GPT models, though its degradation arises from a different failure mode. While its Wasserstein distance is not substantially worse than GPT-4, inspection of the predicted outcome distributions reveals that the model often collapses to deterministic outputs, producing the *same value across all Monte Carlo samples* even when increasing the sampling temperature beyond 1. This lack of distributional diversity

Table 12: *Impact of additional tests on risk estimation.* Mean ± standard deviation of normalized risk estimates as the number of tests increases. Lower values indicate better calibration.

| Dataset | Model | 1 Test | 3 Tests | 5 Tests | All Features |
|---|---|---|---|---|---|
| Kidney | GPT-4o | 0.259 ± 0.007 | 0.267 ± 0.003 | 0.273 ± 0.004 | **0.274 ± 0.002** |
| | GPT-4o-mini | 0.292 ± 0.007 | 0.330 ± 0.010 | 0.349 ± 0.009 | 0.377 ± 0.003 |
| Hepatitis | GPT-4o | 0.209 ± 0.012 | 0.255 ± 0.008 | 0.285 ± 0.018 | 0.326 ± 0.005 |
| | GPT-4o-mini | 0.275 ± 0.019 | 0.307 ± 0.009 | 0.323 ± 0.015 | 0.374 ± 0.003 |
| Diabetes | GPT-4o | 0.575 ± 0.062 | 0.638 ± 0.013 | 0.698 ± 0.096 | 0.687 ± 0.082 |
| | GPT-4o-mini | 0.606 ± 0.014 | 0.696 ± 0.018 | 0.725 ± 0.013 | 0.768 ± 0.004 |

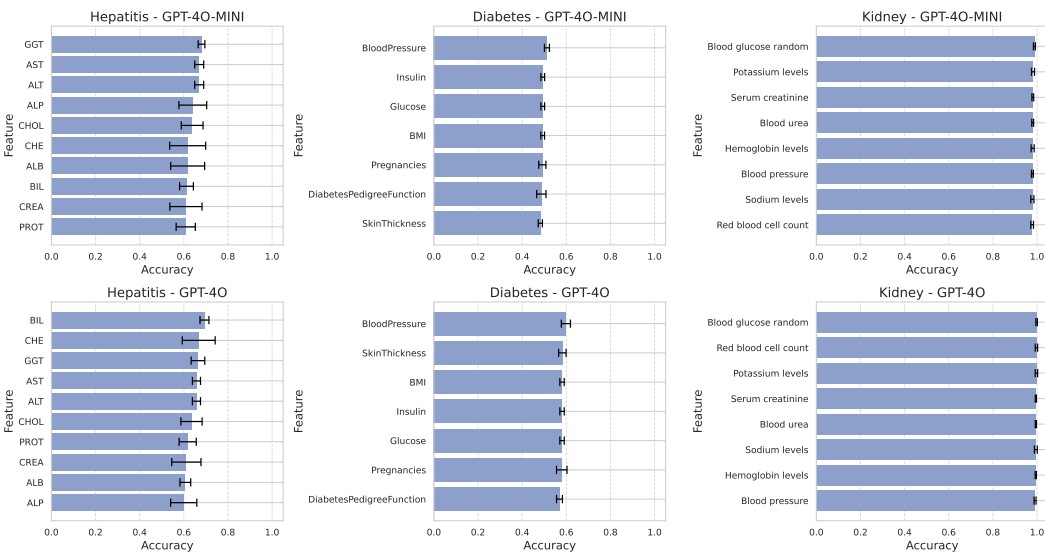

Figure 13: *Feature performance.* Mean accuracy per feature, measured across multiple seeds, for two models (gpt-4o-mini and gpt-4o) on three datasets. Bars represent mean accuracy (0–1), and black error bars denote ±1 standard deviation. Features are ranked in descending order of mean accuracy.

Table 13: *Wasserstein distance between true empirical and model-generated test outcome distributions.* Reported as mean ± standard deviation (lower is better).

| Dataset | Model | Avg. Wasserstein (Mean ± Std) |
|---|---|---|
| Diabetes | GPT-4o | 0.110 ± 0.038 |
| | GPT-4o-mini | 0.117 ± 0.046 |
| | BioMistral-7B | 0.125 ± 0.048 |
| Hepatitis | GPT-4o | 0.130 ± 0.157 |
| | GPT-4o-mini | 0.173 ± 0.237 |
| | BioMistral-7B | 0.425 ± 0.453 |
| Kidney | GPT-4o | 0.082 ± 0.055 |
| | GPT-4o-mini | 0.082 ± 0.057 |
| | BioMistral-7B | 0.215 ± 0.225 |

prevents the model from capturing uncertainty in test outcomes. Consequently, Bayesian Experimental Design (BED) provides no measurable performance improvement, and the overall diagnostic accuracy remains low (see Table 14).

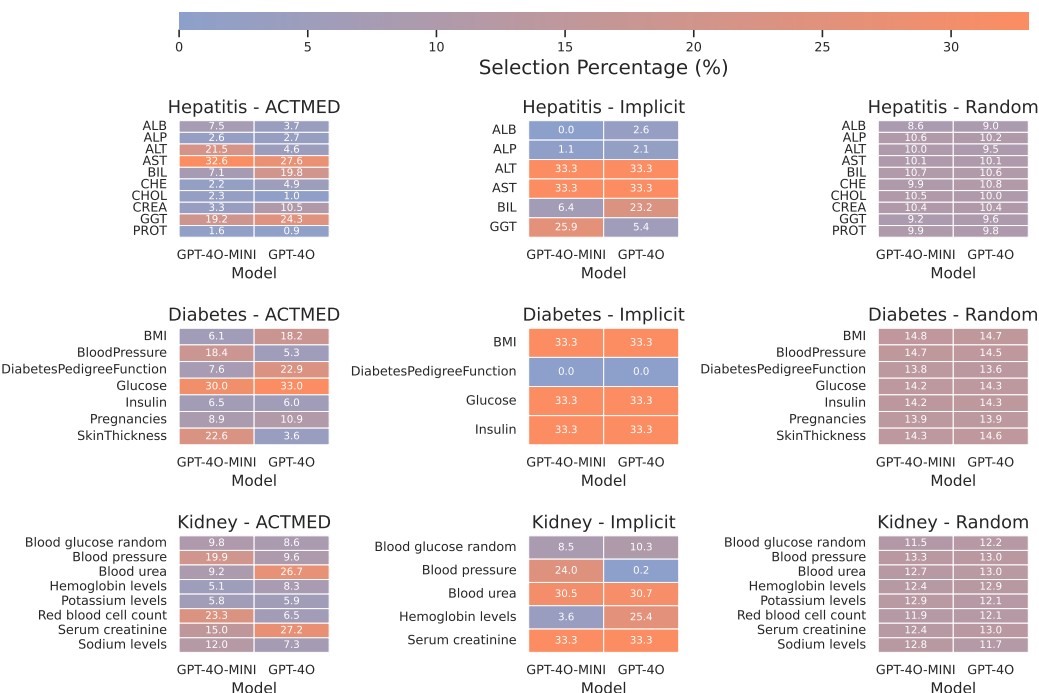

Figure 14: *Feature selection by method.* Heatmaps of feature-selection frequencies (0–100%) across three datasets (hepatitis, diabetes, and kidney) and three selection methods (`ACTMED`, Implicit, Random). Columns represent GPT-4O-Mini and GPT-4O. Each cell shows how often a feature was included across all seeds

Table 14: *Accuracy of different models on the Hepatitis, Diabetes, and Chronic Kidney Disease datasets using all features and `ACTMED`-selected features.* Results are reported as mean $\pm$ standard deviation.

| Dataset | Model | `ACTMED` (Mean $\pm$ Std) | All Features (Mean $\pm$ Std) |
|---|---|---|---|
| **Diabetes** | GPT-4o | $0.682 \pm 0.012$ | $0.584 \pm 0.011$ |
| | GPT-4o-mini | $0.593 \pm 0.039$ | $0.451 \pm 0.062$ |
| | Biomistral-7B | $0.529 \pm 0.025$ | $0.450 \pm 0.008$ |
| | LLaMA-70B | $0.540 \pm 0.013$ | $0.474 \pm 0.012$ |
| **Kidney** | GPT-4o | $0.999 \pm 0.003$ | $1.000 \pm 0.000$ |
| | GPT-4o-mini | $0.992 \pm 0.006$ | $0.975 \pm 0.006$ |
| | Biomistral-7B | $0.657 \pm 0.021$ | $0.564 \pm 0.033$ |
| | LLaMA-70B | $0.984 \pm 0.003$ | $0.986 \pm 0.000$ |
| **Hepatitis** | GPT-4o | $0.839 \pm 0.013$ | $0.807 \pm 0.004$ |
| | GPT-4o-mini | $0.825 \pm 0.013$ | $0.793 \pm 0.009$ |
| | Biomistral-7B | $0.489 \pm 0.029$ | $0.511 \pm 0.027$ |
| | LLaMA-70B | $0.699 \pm 0.023$ | $0.793 \pm 0.010$ |

**Bayesian bootstrap**

To quantify uncertainty in diagnostic predictions, we performed **10 independent diagnostic trials per patient** for each model and dataset. The resulting prediction accuracies were aggregated using **Bayesian bootstrapping** [87],

which provides non-parametric estimates of posterior distributions over mean accuracy values and corresponding **95% credible intervals**. This approach captures both inter-patient and intra-model variability and enables estimation of worst-case performance bounds. Table 15 reports the *mean* and *standard deviation* of predicted risk probabilities, together with lower and upper bounds of the 95% Bayesian bootstrap credible intervals, for each dataset, model, and true disease label. GPT-4o and GPT-4o-mini both show strong stratification between positive and negative classes, with narrower credible intervals in GPT-4o-mini for highly separable diseases (e.g., chronic kidney disease).

Table 15: *Average model prediction accuracy across the Hepatitis, Diabetes, and CKD (Kidney) datasets.* Values show the mean and standard deviation of predicted risk with 95% Bayesian bootstrap credible intervals, stratified by true disease label.

| Dataset | Model | Label | Mean Avg. Risk | Std. Avg. Risk | 95% CI [Lower, Upper] |
|---|---|---|---|---|---|
| Diabetes | GPT-4o | 0 | 0.548 | 0.275 | [0.512, 0.583] |
| | GPT-4o | 1 | 0.799 | 0.143 | [0.782, 0.815] |
| | GPT-4o-mini | 0 | 0.716 | 0.223 | [0.686, 0.744] |
| | GPT-4o-mini | 1 | 0.856 | 0.039 | [0.836, 0.875] |
| Hepatitis | GPT-4o | 0 | 0.093 | 0.129 | [0.081, 0.106] |
| | GPT-4o | 1 | 0.561 | 0.320 | [0.539, 0.583] |
| | GPT-4o-mini | 0 | 0.167 | 0.147 | [0.157, 0.178] |
| | GPT-4o-mini | 1 | 0.581 | 0.289 | [0.555, 0.605] |
| Kidney (CKD) | GPT-4o | 0 | 0.039 | 0.033 | [0.029, 0.051] |
| | GPT-4o | 1 | 0.907 | 0.060 | [0.898, 0.916] |
| | GPT-4o-mini | 0 | 0.179 | 0.119 | [0.161, 0.197] |
| | GPT-4o-mini | 1 | 0.908 | 0.061 | [0.897, 0.917] |

### Example of ACTMED's Test Evaluation Process with Clinician-in-the-Loop

To illustrate ACTMED's information-theoretic test selection process, we present a simplified synthetic binary diagnostic task involving chronic kidney disease (CKD) with two tests. At each step, ACTMED supports clinician decision-making by maintaining transparency over test evaluations and allowing human review.

**Step 1: Prior Belief.** The system starts with a diagnostic prior based on available information:

$$P(y_{d_t} = 1 \mid K_t) = \mathbb{B}(p_{\text{prior}}), \quad \text{where} \quad p_{\text{prior}} = 0.20$$

*Clinician role:* The clinician can inspect the current risk estimate and adjust the prior based on additional context not currently captured in the structured data.

**Step 2: Candidate Test Outcomes.** ACTMED considers two candidate tests and simulates plausible results using a surrogate model:

- **Creatinine (high)** : $2.3, \text{mg/dL}$
- **Creatinine (normal)** : $1.0, \text{mg/dL}$
- **Sodium (normal)** : $140 \, \text{mmol/L}$
- **Sodium (low)** : $130 \, \text{mmol/L}$

*Clinician role:* The clinician may review the simulated outcomes for plausibility, reject irrelevant or infeasible tests, and flag preferred tests based on domain knowledge or patient-specific factors.

**Step 3: Posterior Beliefs.** ACTMED computes updated diagnostic beliefs for each hypothetical outcome:

- **Creatinine**
    - high: $P(y_{d_t} = 1 \mid K_t, \text{high}) = 0.65$
    - normal: $P(y_{d_t} = 1 \mid K_t, \text{normal}) = 0.22$
- **Sodium**
    - low: $P(y_{d_t} = 1 \mid K_t, \text{low}) = 0.45$
    - normal: $P(y_{d_t} = 1 \mid K_t, \text{normal}) = 0.18$

*Clinician role:* The clinician can examine the impact of each test result on the diagnostic belief, and assess whether these posterior shifts are clinically meaningful or likely to influence treatment decisions.

**Step 4: Utility of Each Test.** We compute the expected information gain from each test using the KL divergence between posterior and prior diagnostic beliefs. This utility is expressed as:

$$\mathcal{F}(u_t^{\text{crea}}) = \mathbb{E}[\text{KL}(P(y_{d_t} = 1 \mid K_t, u_t^{\text{crea}}) \parallel P(y_{d_t} = 1 \mid K_t))] = 0.134,$$

$$\mathcal{F}(u_t^{\text{sod}}) = \mathbb{E}[\text{KL}(P(y_{d_t} = 1 \mid K_t, u_t^{\text{sod}}) \parallel P(y_{d_t} = 1 \mid K_t))] = 0.056.$$

Since $\mathcal{F}(u_t^{\text{crea}}) > \mathcal{F}(u_t^{\text{sod}})$, `ACTMED` selects the serum creatinine test as it provides higher expected diagnostic value.

*Clinician role:* Before confirming the selected test, the clinician may override the choice if the utility estimate contradicts clinical judgment, safety concerns, or logistical constraints.

Once a test is acquired, the diagnostic process iteratively continues until a diagnosis is achieved. Figure 15 details how the model supports this by generating intermediate outputs that clinicians can review throughout the process.

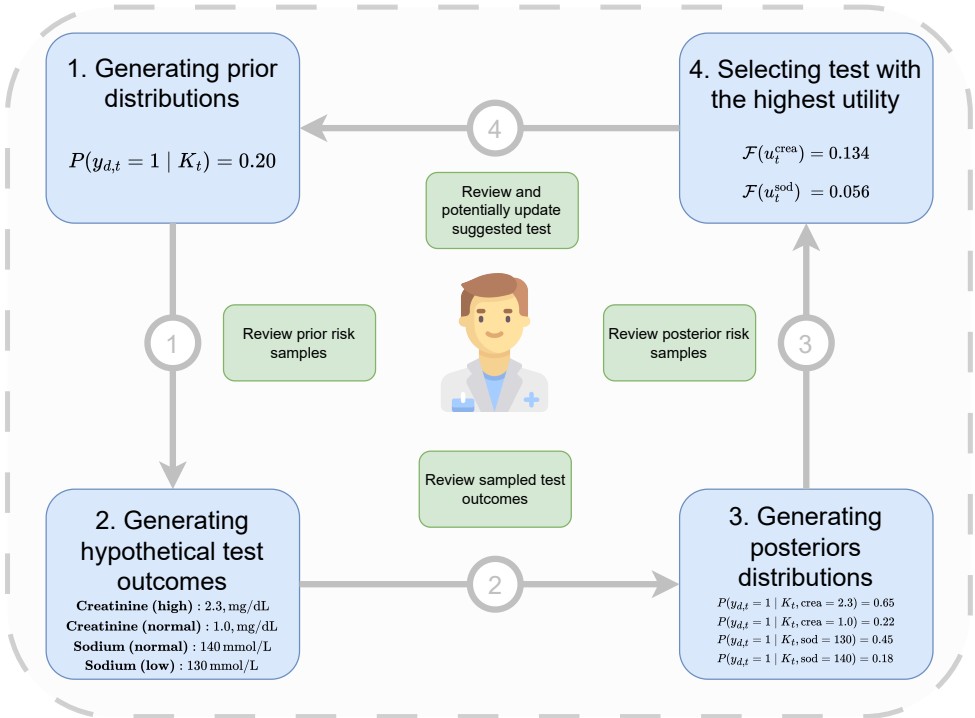

Figure 15: *Illustrative example of* `ACTMED` *'s diagnostic reasoning.* The current belief (prior) about CKD is updated based on hypothetical outcomes for two candidate tests (serum creatinine and sodium). Posterior probabilities differ for each outcome, and the expected KL divergence determines which test offers the greatest diagnostic value. Serum creatinine yields higher expected information gain and is selected.

# F   Computational Cost Analysis

Our framework relies on repeatedly querying a LLM to simulate plausible estimates for candidate diagnostic test results and estimate disease posteriors. Consequently, the computational cost is dominated by the number of LLM queries required per patient episode.

At each decision step $t$, the agent evaluates a set of candidate diagnostic tests $U_t \subset \mathcal{U}_t$, where $|U_t|$ denotes the number of available tests. For each candidate test $u_t^{(i)} \in U_t$, the agent samples $M$ possible outcomes and the resulting hypothetical posterior probability from the LLM to approximate the expected KL divergence. Thus, the computational cost per decision step is $\mathcal{O}(|U_t|M)$ LLM queries.

Let $T$ denote the maximum number of decision steps per patient before either termination or diagnosis. Then, the total computational cost per patient is:

$$\mathcal{C} = \mathcal{O}\left(\sum_{t=1}^{T} |U_t|M\right). \tag{14}$$

In the worst case, where no early termination occurs and all tests are considered at every step, the complexity simplifies to:

$$\mathcal{C} = \mathcal{O}(TNM), \tag{15}$$

where $N$ is the total number of possible diagnostic tests. In practice, $N$ may already be quite small as there will only be a subset of all available tests that can be ordered for diagnosing conditions due to existing guidelines and the model only needs to determine which of those offers the highest utility. Table 16 summarizes the asymptotic computational complexity of our method compared to baseline approaches. While our method incurs higher per-patient computational cost compared to static classifiers, this is justified in domains like clinical medicine where information acquisition is expensive and decision quality is paramount. Inference time depends primarily on LLM latency and hardware availability. In our setup, estimating the expected value of a single diagnostic test takes approximately 20 seconds using either GPT-4o or GPT-4o-mini. Running the full suite of experiments across five random seed, parallelized under a shared API key for each model, takes roughly 60 hours. In deployment, inference time could be significantly reduced through parallelization of the independent API requests. This cost and time delay is negligible relative to the clinical and financial burden of unnecessary or delayed testing. Furthermore, computational demands can be significantly reduced by training lightweight surrogate models specialized for predicting test outcomes, as demonstrated in prior work [88].

Table 16: *Comparison of per-patient computational complexity across methods.*

| Method | Complexity | Description |
|---|---|---|
| Static classifier | $\mathcal{O}(1)$ | Single forward pass using all features |
| Stochastic feature acquisition | $\mathcal{O}(1)$ | Random subset of all features selected |
| Greedy feature acquisition | $\mathcal{O}(T)$ | Selects top-$k$ features in static order |
| **BED with KL divergence (ours)** | $\mathcal{O}(TNM)$ | Actively selects using KL divergence |

