# OpenReview forum: "Timely Clinical Diagnosis through Active Test Selection"
_NeurIPS.cc/2025/Conference — NeurIPS 2025 poster_

### Official Review · Reviewer_JEyC · 2025-06-23

**Clarity:** 3
**Significance:** 2
**Originality:** 3
**Rating:** 3
**Confidence:** 3

**Summary:**

This paper introduces ACTMED, a model-agnostic, adaptive diagnostic framework that supports timely and resource-aware clinical decision-making. The core idea is to integrate Bayesian Experimental Design (BED) with large language models (LLMs) to guide sequential test selection for individual patients. Rather than relying on static, fully observed datasets, ACTMED mimics how clinicians reason over time—prioritizing tests that are expected to reduce diagnostic uncertainty the most (see Figure 2 of the paper).
At each step, ACTMED uses LLMs (e.g., GPT-4o) as surrogate models to simulate plausible patient test outcomes and to estimate the diagnostic utility of candidate tests via expected KL divergence. A stopping rule is introduced to avoid redundant testing, based on whether further tests are expected to meaningfully shift diagnostic beliefs.

**Questions:**

1. Why does the proposed method outperform models that use the full set of features in the simulation?
2. How does the method address potential overfitting, especially given the small sample sizes used in evaluation?
3. How does this approach compare to a pipeline where LLMs are used for imputing missing features, followed by conventional classification models?
4. In the binary diagnostic setting, how are the “true” diagnostic probability distributions defined and validated?
5. How does the proposed framework scale to and perform on larger, more complex clinical datasets such as MIMIC-III/IV, UK Biobank, or All of Us?
6. How would the clinician-in-the-loop component operate in practice—can clinicians override the model’s test selection, and is this interaction supported in the current framework?

**Ethical Concerns:**

["NO or VERY MINOR ethics concerns only"]

**Final Justification:**

I appreciate the authors' efforts and have increased my score to 3. However, the concern remains: the statement that “scaling ACTMED to very large clinical datasets is challenging” raises questions about whether the reported performance gains may be due to overfitting.

**Limitations:**

Yes.

**Quality:**

2

**Strengths And Weaknesses:**

Strengths
1. The paper addresses a well-motivated and clinically important problem: how to make timely diagnoses in a cost-effective manner.
2. The use of large language models (LLMs) as a flexible sampling engine for test selection is an interesting/potentially novel idea.

Weaknesses
1. The diagnosis task is limited to binary classification. It is unclear where the “true” diagnostic probability distributions come from, especially given the reliance on LLM-generated samples.
2. The method is framed as an agent-based decision process, but it lacks comparison to simpler alternatives such as using LLMs for missing data imputation followed by standard prediction.
3. The paper claims clinicians can be kept in the loop, but this is not demonstrated or evaluated in the experiments.
4. The real-world examples used for validation (e.g., hepatitis, diabetes, CKD datasets) are small and outdated, with only ~100 patients and ~20 covariates. This limits the assessment of scalability and clinical relevance.

---

> ### Author Rebuttal · Authors · 2025-07-30
>
> *We thank the reviewer for their constructive comments. We have addressed the reviewer’s concerns below:*
>
> ---
>
> ## **Binary classification and diagnostic probability distributions**
> Binary classification in a one vs all manner is a common approach in medicine once a differential diagnosis list has been established for test selection (see response to Reviewer tmUM). ACTMED does not assume access to a ground-truth diagnostic probability distribution. Instead, it leverages LLM-generated samples to approximate plausible posteriors under different hypothetical test outcomes. This does mean performance depends on the accuracy of the LLM generated samples and probability estimates (see response to Reviewer tmUM), but we demonstrate that this approach improves downstream classification accuracy. The test that most meaningfully shifts the model’s disease probability estimate is recommended. Final predictions are then evaluated against ground truth labels, allowing comparison of test selection quality across methods.
>
> ---
>
> ## **Comparison to LLM-based imputation pipelines**
> While pipelines that combine LLM based imputation with classifiers (1) can work well, they typically require task specific labelled training data, risk introducing distributional biases, and do not support sequential, adaptive feature acquisition. ACTMED is designed to operate zero shot and focuses on optimizing which tests to perform rather than replacing clinician driven diagnosis. We have added a more detailed discussion of imputation methods, including those using LLMs, in the related work section.
>
> ---
>
> ## **Clinician-in-the-loop**
> We have now included a qualitative evaluation with experienced clinicians and senior medical students, who reviewed ACTMED’s reasoning pathways (see response to reviewer tmUM). Clinicians valued ACTMED’s ability to suggest tests when they were uncertain, its intuitive stopping rule, and its supportive nature. As detailed in the study, clinicians can examine the intermediate outputs and assess how the model is changing its risk in response to the lab value changes. Clinicians can intervene at three points: by adjusting the predicted lab values, modifying the assigned risk estimate, or overriding the final test recommendation. In practice, reviewing distributions and risk changes can be time consuming, so most clinicians under time pressure would focus primarily on evaluating the suggested test. After incorporating the clinician’s input, whether it is confirming the suggested test or selecting a different one, ACTMED updates its predictions and proceeds accordingly.
>
> ---
>
> ## **On Dataset Scale and Realism**
> Scaling ACTMED to very large clinical datasets is challenging due to (a) privacy restrictions on LLM usage and (b) the fact that many datasets only record a single test trajectory per patient, limiting their suitability for sequential testing studies. We address this by dynamically constructing multiple synthetic trajectories from tabular data to generate EHR style notes as tests are selected.
> We emphasize that the cited datasets contained roughly 100 patients only but this was the size of the test set used in the model evaluation. ACTMED is not subject to overfitting in the conventional sense, as it performs no fine-tuning and does not update its parameters based on the evaluation data. Its zero-shot design ensures it avoids learning dataset-specific shortcuts. In contrast, baselines that rely on full-feature classification are more prone to overfitting, especially when operating on small, fully observed datasets. While the datasets are older, the clinical challenges and measurements remain relevant, as discussed in Section 4.
>
> To strengthen the evaluation, we extended the diabetes analysis to include all 768 patients (Table 1) and added a more complex evaluation on an AgentClinic-derived dataset (see response to Reviewer fFNv). Together, these bring the total number of evaluated patients to over 1,000 and cover a wider range of covariates.
>
> **Table 1:** *Performance of GPT-4o and GPT-4o-mini on the full diabetes dataset (n=768) against the baselines.*
> | Model       | Method         | Mean   | Std   |
> |-------------|----------------|--------|-------|
> | GPT-4o-mini | ACTMED         | 0.593  | 0.039 |
> | GPT-4o-mini | Random         | 0.505  | 0.027 |
> | GPT-4o-mini | Global Best    | 0.485  | 0.042 |
> | GPT-4o-mini | Implicit       | 0.488  | 0.040 |
> | GPT-4o-mini | All Features   | 0.451  | 0.062 |
> | GPT-4o      | ACTMED         | 0.682  | 0.012 |
> | GPT-4o      | Random         | 0.565  | 0.014 |
> | GPT-4o      | Global Best    | 0.540  | 0.008 |
> | GPT-4o      | Implicit       | 0.544  | 0.004 |
> | GPT-4o      | All Features   | 0.584  | 0.011 |
>
> ---
>
> ## **References**
> 1.	Active Learning with LLMs for Partially Observed and Cost-Aware Scenarios (Astorga et al. NeurIPS 2024)
>
> ---
>
> *Thank you for your thoughtful feedback. We appreciate your insights, which have helped improve the quality and clarity of our work. We believe ACTMED provides a practical step toward generalizable, interpretable AI in clinical diagnostics, and we welcome continued dialogue.*

---

> > ### Comment · Reviewer_JEyC · 2025-08-05
> >
> > I appreciate the authors' efforts and have increased my score. However, the concern remains: the statement that “scaling ACTMED to very large clinical datasets is challenging” raises questions about whether the reported performance gains may be due to overfitting.

---

> > > ### Author Response · Authors · 2025-08-05
> > >
> > > Dear Reviewer,
> > >
> > > Thank you for your continued engagement and for raising your score. Below we clarify why ACTMED’s reported gains are **not** due to over-fitting and why large-scale datasets pose mainly *structural*—not statistical—challenges for evaluation.
> > >
> > > We respectfully note that “over-fitting” is usually reserved for supervised-learning scenarios—when a model trained on labelled data attains high training accuracy but degrades on unseen test data (1). Because ACTMED involves no training on the evaluation datasets and is evaluated in a fully zero-shot setting with general-purpose LLMs, we believe “over-fitting” is not the appropriate term here.
> > >
> > > There are several concrete reasons the observed performance gains on the tested datasets are extremely unlikely to be spurious artifacts:
> > >
> > > ---
> > >
> > > ### 1. No task-specific training or prompt tuning
> > > * **Zero fine-tuning** ACTMED never trains a classifier nor adapts the LLM to any dataset.
> > > * **Single universal prompt** Exactly the same prompt template is used for every patient and across baselines. The full prompt text is given in the Appendix.
> > > * **Implication** All performance gains come from *better test selection*, not model fitting.
> > >
> > > ---
> > >
> > > ### 2. Genuine zero-shot feature acquisition
> > > * **Selection mechanism** The LLM samples hypothetical test outcomes; tests are ranked by expected KL-divergence reduction.
> > > * **Risk of label leakage** Like any LLM, pre-training could incidentally memorise snippets of public datasets, but this would affect all methods equally and not favour ACTMED.
> > > * **Data-contamination risk** If spurious correlations in feature selection due to data leakage were inflating results, they would appear more strongly in the **Global-Best** and **Implicit** baselines, which read the entire feature set. ACTMED merely produces risk estimates given hypothetical changes in information; the KL score alone decides which test to pick.
> > >
> > > ---
> > >
> > > ### 3. Why “very large” datasets remain difficult to *evaluate*
> > > * **Structural mismatch** Datasets such as *MIMIC-IV* record a single fixed test trajectory per patient; they lack the multiple partial states required to benchmark sequential test policies. To our knowledge there are no large public datasets that focus on sequential diagnostic test acqusition explicitly rather than an evaluation of LLMs performance in clinical diagnosis.
> > > * **Additional benchmarks confirm performance gains** To compensate, we built an **AgentClinic-derived** dataset with richer trajectories, and ACTMED’s gains persist. Extending the diabetes experiment to the full 768-patient cohort (same prompts, same models) produced the same performance trend.
> > >
> > > ---
> > >
> > > **Reference**
> > > 1. An Overview of Overfitting and its Solutions*, (Xing Journal of Physics, 2022)
> > >
> > > ---
> > >
> > > We hope this clarifies why ACTMED’s improvements over baselines cannot be attributed to supervised-learning over-fitting. Please let us know if further clarification would be helpful.
> > >
> > > Best regards,
> > > The authors of submission 22995

---

> > > > ### Author Response · Authors · 2025-08-09
> > > >
> > > > Dear Reviewer,
> > > >
> > > > Thank you very much for taking the time to thoughtfully consider our responses and for your constructive engagement throughout the process. We greatly appreciate your recognition of the contributions and improvements in the revised submission.
> > > >
> > > > Best wishes,
> > > >
> > > > The authors of submission 22995

---

### Official Review · Reviewer_pTAQ · 2025-07-02

**Clarity:** 2
**Significance:** 3
**Originality:** 2
**Rating:** 4
**Confidence:** 4

**Summary:**

This paper casts clinical diagnosis as a sequential decision making problem: where every step is an action of selecting the next diagnostic test with the aim of diaganosing the patient successfully. To achieve this, they propose ACTMED (Adaptive Clinical Test selection via Model-based Experimental Design) - this method combines Bayesian Experimental Design (BED) with large language models. LLMs are used to (1) simulate plausible test outcomes based on the available information so-far and (2) the disease-probability distribution given the information so far, effectively turning LLMs into on-demand generative priors.Tests are chose based on expected KL div between prior distribution and posterior distribution if a test is conducted (normaized by their costs). This procedural loop continues until no remaining test can meaningfully refine the belief, after which a clinician can inspect, accept, or override the transparent intermediate reasoning—keeping humans firmly “in the loop.” Experiments on three real-world tabular datasets show that ACTMED delivers higher diagnostic accuracy than baseline strategies, cuts the average number of tests, demonstrating gains in accuracy, interpretability, and resource use simultaneously.

**Questions:**

**Major comments**

(1) Please correct or explain (2) and (3)

(2) It would be helpful to justify the use of KL divergence (as opposed to entropy criteria) as a selection criteria by some concrete theoretical proof and experiment.

(3) How is the LLM calibration (some evaluation on that), and how it impacts performance would be helpful in more concretely bear out the impact underlying LLM accuracy on algorithm's performance.

(4) Improve the writing of Section 2,3 and 4.1




**Minor comments**

(1) Line 142 - Is it possible that prior is not confident enough and none of the test satisfied condition presented in Line 142 - what to do in that case?

(2) Why using all features is worst (Figure 5) is not entirely clear to me - why this should be the case? What if we ask ACTMED do sequentially sample all the features (without early stopping) - will its performance (accuracy) start decreasing after some steps?

**Ethical Concerns:**

["NO or VERY MINOR ethics concerns only"]

**Final Justification:**

The authors have addressed most of my concerns and added additional experiments to support it. The paper’s writing and exposition still needs some refinement, I am raising my score by 1 point to weak accept.

**Limitations:**

Yes there is a dedicated section on the Limitations of the paper.

**Paper Formatting Concerns:**

No major concerns regarding formatting of the paper.

**Quality:**

2

**Strengths And Weaknesses:**

**STRENGTHS**

(1) **High-impact, well-motivated problem** – frames diseases diagnosis as personalized, cost-aware decision-making problem that promises earlier detection, lower expenditure, and wider access to care.

(2) **Flexible framework** –  Model agnostic and runs with any without task-specific fine-tuning, suggesting easy portability (although performance depends on the underlying LLM accuracy) ; allows for clinician to be in loop - simulated lab values and belief updates are readable, so experts can audit or override recommendations..

(3) **Thorough empirical study** – on three public datasets, two different models, and comparisons against guideline-style, random, implicit-LLM, and full-feature baseline (although I have concerns with some of them - please see questions section). Additionally Section 4.3 is an interesting read.


**WEAKENESSES**

(1) **Notational inconsistency** - following terms are not clearly defined and are hard to read - prior, posterior over ''disease probability`` are not clearly defined and are used interchangeably with prior and posterior over ''disease label''  (which conflates epistemic and aleatoric uncertainty). Similarly for the distribution of test outcomes and other quantities.

(2) **Equation (2) seems incorrect** - Expectation should be over different posteriors (due to different test outcomes). KL divergence should be between prior and posterior of the ``disease probability''. And not on samples basis as done in Equation (2). Explained below: Let $\theta$ be the probability that person has disease or not ($Y=\{0,1\}$), that is $Y \sim p_\theta$. And we have a prior $\pi(\theta|K_t)$ on $\theta$ at time step $t$ given information $K_t$. Overall $P(Y) =\int p_\theta (Y) \pi(\theta|K_t) d\theta$. Now, the test outcome $u_t^{i} \sim P(u_t^{i}|K_t)$, then we are intersted in
$$ E_{u_t^{i} \sim P(u_t^{i}|K_t)} KL(\pi(\theta|K_t) || \pi(\theta|K_t,u_t^{i}))$$ which can be approximate by
$ \frac{1}{K} \sum_{j=1}^K KL(\pi(\theta|K_t) || \pi(\theta|K_t,u_t^{i,j}))$.

Now, $KL(\pi(\theta|K_t) || \pi(\theta|K_t,u_t^{i,j}))$ will depend on distributions $\pi(\theta|K_t)$ and $\pi(\theta|K_t,u_t^{i,j})$ which are not bernoulli but something else.

(3) Rationale behind Equation (3) is not given.

(4) Applications are limited to tasks on which GPT-4o performs good - it seems for other tasks finetuning would be crucial.

(5) Overall writing and presentation needs improvement - especially Sections 2 and 3, Subsection 4.1

---

> ### Author Rebuttal · Authors · 2025-07-30
>
> *We thank the reviewer for their helpful feedback, particularly on enhancing the clarity of the mathematical formulation.*
>
> ---
>
> ## **Formulation of the BED using KL divergence**
>
> We now clearly distinguish between priors and posteriors over *disease probabilities* versus *disease labels*, and we clarify that the KL divergence in Eq.​(2) is computed between *probability* distributions (not between the labels themselves).
>
> ### 1. Epistemic vs. Aleatoric in Active Learning
>
> In Active Learning (common BED framework), we aim to reduce the epistemic uncertainty by acquiring the label of the most informative sample $x_o$, which.
>
> $\underbrace{I(Y;Z\mid, x_o, d)}_{\text{epistemic}} = H(Y\mid d, x_o)-   \underbrace{H(Y\mid Z,d, x_o)}\_{\text{aleatoric}} ,$
>
> where $Z$ is a R.V. representing the latent hypothesis (e.g. parameters distribution), $d$ denotes current data, $H(\cdot)$ is entropy and $I(\cdot)$ mutual information.
> * **Epistemic** $=$predictive $-$aleatoric
>
> ---
>
> ### 2. Active Feature Acquisition (AFA)
>
> Our setting is closer to Active Feature Acquisition, where we ask how much a candidate feature $F_j$ reduces **predictive** uncertainty in $Y$:
>
> $\displaystyle I(Y;F_j)=H(Y)-H(Y\mid F_j).$
>
> Because $H(Y)$ is constant across features, maximizing $I(Y;F_j)$ is equivalent to minimizing $H(Y\mid F_j)$; the most informative test is the one that lowers entropy most.
>
> ---
>
> ### 3. Our KL‑Based Criterion and Its Derivation
>
> When computing expected information gain we may work interchangeably in entropy or KL‑divergence space (1).  Equation (2) evaluates
>
> $E[\mathrm{KL}] = -H(Y \mid F_j) + \mathrm{CE}\bigl(Y, Y_{\text{prior}}\bigr).$
>
>
> where
> * **Term A** ($-H(Y\mid F_j)$) is exactly the uncertainty‑reduction term in AFA.
> * **Term B** is a cross‑entropy that rewards tests which move beliefs away from an uninformative prior, thus avoiding redundant queries.
>
> > **Note on notation:** the prior probability $p_{\text{prior}}$ is constant and no longer carries an index $j$, this has been corrected in the manuscript.
>
> #### *Concrete derivation*
> For a single Monte‑Carlo sample we compute
>
> $p_{\text{post}}\log \frac{p_{\text{post}}}{p_{\text{prior}}}+ (1-p_{\text{post}})\log \frac{1-p_{\text{post}}}{1-p_{\text{prior}}}$
>
> Re‑arranging,
>
> $= \bigl[p_{\text{post}}\log p_{\text{post}} +(1-p_{\text{post}})\log(1-p_{\text{post}})\bigr]$
> $- \bigl[p_{\text{post}}\log p_{\text{prior}} +(1-p_{\text{post}})\log(1-p_{\text{prior}})\bigr]$
>
> Where the term in the first brackets corresponds to $A$ and the second to term $B$. Averaging over $M$ such samples yields the expectation in Eq.​(2). The full derivation has now been included in Appendix B.
>
> ---
>
> ## **Clarity on the utility function notation**
> First, **Equation (1) is now expressed in standard minimisation form** as suggested by reviewer fFNv:
>
> $\mathcal{L}(y,u_t,\lambda)=\sum_t \mathbb{I}[y\neq y_{d_t}]+\lambda\sum_t c(u_t)$
>
> Second, **to prevent symbol overload we separate the random variable from its realisation**:
> throughout Section 2 (lines 110–114), $u_t$ denotes the *type* of diagnostic test (a random variable), while $u_t'$ denotes the *observed outcome* of that test.
>
> We relax the dual objective with
> $\mathcal{F}(u_t^i)=\frac{I(u_t^i)}{c^*(u_t^i)},$
>
> taking
> $c(u)=\log c^{u}$
> to model diminishing marginal penalty.  All experiments assume unit costs
> $(c^*=1)$
> so baselines remain directly comparable; real monetary or invasiveness costs can be inserted transparently.  Normalizing information gain by cost gives an intuitive “information‑per‑dollar” score and removes the need to hand‑tune a dataset‑specific $\lambda$. The gain per unit cost formulation is also common in medicine (2).
>
> ## **Use of KL Divergence versus Entropy for BED**
>
> In Appendix B, we present a theoretical example illustrating how an entropy-based criterion can fail to identify the most informative test, specifically in cases where entropy is high but concentrated on outcomes that provide little diagnostic value. Both entropy and KL divergence can, in principle, be used for test selection (1), and there is no universal proof favoring one over the other; their relative effectiveness depends on the specific setting and data distribution. However, in our experiments, we provide both a theoretical justification and empirical evidence supporting KL divergence for our application. Across all datasets, the results (Table 1) demonstrate that KL-based selection consistently improves diagnostic accuracy and reduces redundant testing relative to entropy-based selection.
>
> **Table 1:** *Performance of ACTMED using KL divergence versus entropy for test selection.*
>
> | Model       | Dataset   | KL-based (Mean ± Std) | Entropy-based (Mean ± Std) |
> |-------------|-----------|------------------------|-----------------------------|
> | GPT-4o-mini | Diabetes  | 0.593 ± 0.039         | 0.572 ± 0.005              |
> | GPT-4o-mini | Kidney    | 0.992 ± 0.006         | 0.972 ± 0.010              |
> | GPT-4o-mini | Hepatitis | 0.825 ± 0.013         | 0.770 ± 0.010              |
> | GPT-4o      | Diabetes  | 0.682 ± 0.012         | 0.673 ± 0.011              |
> | GPT-4o      | Kidney    | 0.999 ± 0.003         | 0.994 ± 0.000              |
> | GPT-4o      | Hepatitis | 0.839 ± 0.013         | 0.770 ± 0.017              |
>
> ---
>
> ## **Performance Degradation with All Features**
>
> We confirm that ACTMED’s accuracy can decline when too many features are presented at once. Following the reviewer’s suggestion, we added an experiment examining performance as features are added sequentially (Table 2).
>
> The results show that ACTMED avoids this “information overload” by stopping early when further tests offer little diagnostic value and only selecting informative tests. Similar degradation with excessive input has been observed in prior work in different contexts (3, 4).
>
> One contributing factor is that the models tend to assume higher disease risk as more lab results are presented, even when results are normal (Table 3). This bias is especially evident with GPT‑4o mini.
>
> ---
>
> **Table 2:** *Accuracy versus Number of Sequentially Added Tests*
>
> | Dataset   | Model        | 1 Test            | 3 Tests           | 5 Tests           | All Tests         |
> |-----------|--------------|--------------------|--------------------|--------------------|--------------------|
> | Kidney    | GPT‑4o       | 0.994 ± 0.004     | 0.995 ± 0.003     | 0.995 ± 0.003     | 1.000 ± 0.000     |
> |           | GPT‑4o‑mini  | 0.975 ± 0.011     | 0.983 ± 0.005     | 0.990 ± 0.010     | 0.972 ± 0.006     |
> | Hepatitis | GPT‑4o       | 0.589 ± 0.023     | 0.679 ± 0.022     | 0.718 ± 0.013     | 0.796 ± 0.010     |
> |           | GPT‑4o‑mini  | 0.588 ± 0.030     | 0.698 ± 0.022     | 0.732 ± 0.010     | 0.798 ± 0.011     |
> | Diabetes  | GPT‑4o       | 0.730 ± 0.138     | 0.662 ± 0.170     | 0.664 ± 0.170     | 0.678 ± 0.161     |
> |           | GPT‑4o‑mini  | 0.546 ± 0.015     | 0.524 ± 0.025     | 0.528 ± 0.016     | 0.458 ± 0.007     |
>
> ---
>
> **Table 3:** *Impact of Additional Tests on Risk Estimation*
>
> | Dataset   | Model        | 1 Test            | 3 Tests           | 5 Tests           | All Features      |
> |-----------|--------------|--------------------|--------------------|--------------------|--------------------|
> | Kidney    | GPT‑4o       | 0.259 ± 0.007     | 0.267 ± 0.003     | 0.273 ± 0.004     | 0.274 ± 0.002     |
> |           | GPT‑4o‑mini  | 0.292 ± 0.007     | 0.330 ± 0.010     | 0.349 ± 0.009     | 0.377 ± 0.003     |
> | Hepatitis | GPT‑4o       | 0.209 ± 0.012     | 0.255 ± 0.008     | 0.285 ± 0.018     | 0.326 ± 0.005     |
> |           | GPT‑4o‑mini  | 0.275 ± 0.019     | 0.307 ± 0.009     | 0.323 ± 0.015     | 0.374 ± 0.003     |
> | Diabetes  | GPT‑4o       | 0.575 ± 0.062     | 0.638 ± 0.013     | 0.698 ± 0.096     | 0.687 ± 0.082     |
> |           | GPT‑4o‑mini  | 0.606 ± 0.014     | 0.696 ± 0.018     | 0.725 ± 0.013     | 0.768 ± 0.004     |
>
> ---
>
> ## **LLM Performance within ACTMED**
>
> We agree that the performance of ACTMED depends on the underlying LLM’s ability to generate plausible test outcomes. As noted in our response to Reviewer tmUM, a 7B parameter Biomistral model produced unrealistic samples, limiting ACTMED’s gains. While fine‑tuning could improve small models for specific tasks, our results suggest that ACTMED benefits most from larger, general models capable of diverse, accurate sampling.
>
> ---
>
> ## **Test Stopping Criterion**
>
> If no test meets the KL divergence threshold, ACTMED stops acquiring tests. This reflects many clinical situations, where a confident diagnosis can be made based on history and presentation alone (often referred to as diagnosed clinically by doctors).
>
> In our experiments, this situation was rare, as each patient was provided with only basic demographic and symptom information initially, so additional testing was typically required. However, in principle it would be possible for the model to diagnose a patient “clinically” without requesting further tests.
>
> ---
>
> ## **Writing and Presentation Improvements**
>
> Sections 2 and 3 have been revised to include the more consistent notation. Section 4.1 has also been changed to incorporate the distribution analysis metrics as suggested by Reviewer fFNv. Figures 1 and 2 have been simplified and re-captioned as requested by Reviewer fFNv. Figures 3 and 4 now reflect distributional evaluation metrics.
>
> ---
>
> ## **References**
>
> 1. Modern Bayesian Experimental Design (Rainforth et al. Statist. Sci. 2024)
> 2. Cost-effectiveness Thresholds Used by Study Authors, 1990-2021 (Neumann  et al. JAMA 2023)
> 3. Large Language Models Can Be Easily Distracted by Irrelevant Context (Shi et al., ICML 2023)
> 4. Inverse Scaling in Test-Time Compute (Gema et al., arXiv 2025)
>
> ---
>
> *We appreciate the time and effort you have dedicated to reviewing our paper and hope these additions and clarifications address your concerns satisfactorily. We’d be happy to engage in further discussions.*

---

> > ### Comment · Reviewer_pTAQ · 2025-08-05
> >
> > Thank you for clarifying my questions and conducting the additional experiments.
> >
> > The KL-divergence criterion is now clear to me. Earlier, I got confused due to the notation—it wasn’t clear which KL-divergence the authors meant:
> >
> > (1) KL [ distirbution of Y (after acquiring a feature) || distirbution of Y (before acquiring a feature)  ]
> >
> > or
> >
> > (2) KL [ dist. of disease probability (after acquiring a feature) || dist. of disease probability (before acquiring a feature) ]
> >
> > It would help to state explicitly that you are using (1), not (2), since the terminology overlaps with Bayesian active learning and might be confusing.
> >
> > I will raise my score by 1.

---

> > > ### Author Response · Authors · 2025-08-05
> > >
> > > Dear Reviewer,
> > >
> > > Thank you for the helpful clarification and for raising your score. We’ll make sure the manuscript states explicitly which KL-divergence is used so the distinction is unambiguous.
> > >
> > > We appreciate your constructive discussion throughout the process.
> > >
> > > Best regards,
> > > The authors of submission 22995

---

### Official Review · Reviewer_fFNv · 2025-07-02

**Clarity:** 2
**Significance:** 3
**Originality:** 3
**Rating:** 5
**Confidence:** 3

**Summary:**

The paper proposes ACTMED, a step-wise diagnostic reasoning framework combining LLMs generations with Bayesian Experimental Design / Informativeness of actions.

ACTMED proposes to take a probabilistic view on diagnostic reasoning and to consider actions (i.e. further diagnostic tests) only if they promise to shift the posterior over diagnoses. This way a sequential sampling over most informative tests is implemented.

Overall, this paper has a strong clinical motivation - in clinical decision making we need to carefully select diagnostic actions that provide most insights and have the potential to reduce uncertainty.

Experiments are quite convincing, even if conducted on maybe a bit simplified tasks.

Overall the paper adds value to the community in my view.

**Questions:**

- Eq 1 has a in my view weird notation under the "min", i would ask to use more standard notation
- Fig 1 is poorly explained what is shown, it feels quite out of context, and I don't know what it is trying to tell me (and what exactly the distribution is trying to show)
- Fig 2 again has a bit cluttery and unclear notation in the green boxes on the right (both step 2 and 3).
- l.112 it's a bit unfortunate to use u_t with different upper indices to indicate both the test and its value/outcome --> this makes the notation a bit overloaded and hard to read - i would disentangle that or use RV vs realization lingo
- How does this approach compare to general medical experts / clinicians in solving diagnostic tasks based on same / limited input information?
- I see that lagrange formulation was used for weighing informativeness vs cost, but wouldn't it make sense to actually tune the lambda between those constraints? (Maybe i misunderstood this)
- Please show how and how accurately the LLM predicts the lab test results.
- Fig 3: "Mean predictive performance" -> i have no clue what is shown and meant here.
- For surrogate sampling evaluation: pls explain why this minimal MAE / distance approach is justified (as i believe it will go down automatically with enough samples drawn) --> instead distributional metrics would be more interesting
- l.206 Performance improves with more samples --> this is exactly trivially so if you consider minimal distance to closest generated sample
- l 227 what is "all features" exactly? explain in detail
- what does ACTMED perf > all features mean exactly? What's the mechanism? Which additional info in all features is hurtful and why?
- Reader wants dataset details -> ref to App C when introducing tasks in Sec 4.2
- it's not clear how the "stop testing and go for final diagnosis" procedure is implemented -> e.g. prior works like AgentClinic let an agent state "request test" vs "arrive at final diagnosis". How this is solved here is not so clear
- Fig 7 what does it mean that implicit method is better than proposed? How to interpret this?
- Where are CKD results? Fig 5 and 7 only report Hepatitis and Diabetes
- l 352 says framework allows for clinicians to override suggestions: has this been tested with few clincians? This would strengthen the paper a lot
- l 354 this paragraph sounds a bit ChatGPT written, please reprhase.
- Tab 3: ML and DL column are not very clear what is meant: Machine learning and Deep learning? pls explain more clearly the reasoning in caption
- l 259: Diabetes diagnosis is made (gold standard) with Oral Glucose Tolerance Test (OGTT), not BMI..

**Ethical Concerns:**

["NO or VERY MINOR ethics concerns only"]

**Final Justification:**

Overall, I think the paper can be accepted. It adds a intuitive and interesting perspective of applying Bayesian experimental design / decision making to LLM-based diagnosis. During rebuttal my concerns were adressed, e.g. around "percentage in reasonable range".

**Limitations:**

-Interaction with human clinician is not properly described.
- i don't expect negative societal impact

**Quality:**

3

**Strengths And Weaknesses:**

Strengths:
- Probabilistic reasoning is currently underemphasized by the communtiy in the medical LLM space
- BED makes a lot of sense to use here
- The KL-based loss is an elegant way to determine the value of diagnostic tests
- The framework also allows for modeling of costs of individual tests, which makes it more amenable for deployment.

Weaknesses:
- The experimental part is not super strong: three diagnostic benchmarks are used that are not very typical benchmarks in the medical reasoning / medical LLM community, but rather look like UCI and kaggle benchmarks - which raises the question: why these 3 exactly? (e.g. how do we know that not 6 were tried and 3 reported etc.) Just for sake of comparing to prior literature, it would have been easy to leverage more typical medical reasoning benchmarks such as MedQA, MedExpertQA, AgentClinic etc.
- One aspect that is not clear to me: to which degree is this KL-based formulation for selecting informative tests novel? I haven't seen this done before with LLMs in medical context, but KL and information theoretic aspects seem quite straight forward, so what exactly is the novelty / most similar related work here?
- It is not clear to me how costs are modeled? (i.e. simply $ cost of lab test, which would be oversimplifying, as there are many types of costs to diagnostic tests: invasiveness / health risk introduced by test/intervention (e.g. sampling amniotic fluid), opportunistic cost in triage: who will get into CT first in the rural emergency room?
- Humans are trained to do this sequential diagnostic reasoning, but no comparison to a human baseline is provided to my knowledge: this would strengthen the paper and support the claims made.
- I do not trust the surrogate sampling evaluation metric: predicting lab tests from other tests should be somewhat doable, but I believe it's misleading to report it to be 99% within reasonable range: to evaluate whether a predicted test is reasonable one should report distributional metrics (distribution matching, wasserstein dist etc.) - not the minimal distance to samples (as this flawed metric will trivially degenerate to 0 if enough samples are drawn for 2 distributions with joint support, to my understanding).

---

> ### Author Rebuttal · Authors · 2025-07-25
>
> *We thank the reviewer for their time reviewing our work and their constructive comments.*
>
> ---
>
> ## **Choice of datasets**
> The chosen datasets were selected using strict criteria:
> 1. Open source (to avoid restrictions on LLM use and ensure reproducibility).
> 2. Primarily numerical features (as sampling narrow categorical features offers little benefit from LLM reasoning).
> 3. Well-defined clinical meaning and units, enabling realistic simulation.
> 4. The datasets intentionally do not include a single definitive diagnostic test (e.g., a PCR test for COVID-19), as such features would trivialize the decision-making task.
>
> While these differ from typical benchmarks such as MedQA or AgentClinic, few open-source datasets meet these criteria and support dynamic, sequential test acquisition. Traditional benchmarks generally lack the necessary structure for sequential testing, as ACTMED is designed for the second step of diagnosis (optimal test selection once differentials are known) rather than the initial generation of differentials (see response to reviewer tmUM).
>
> To address concerns about simplicity, we created a dataset derived from MedQA, selecting 120 patients with sufficient test results. For each, we provide a brief EHR-style summary, including synthetic controls with normal lab values, and compare ACTMED’s performance against baselines. Results (Table 1) show ACTMED achieves performance gains across varied diseases, though differing test panels limit detailed population-level sampling analysis.
>
> **Table 1:** *Performance of ACTMED on the AgentClinic derived dataset against baselines.*
>
> | Model      | Method         | Mean Accuracy | Std. Dev. |
> |------------|----------------|---------------|-----------|
> | GPT‑4o‑mini | ACTMED         | 0.697         | 0.015     |
> |            | Random         | 0.596         | 0.015     |
> |            | Global Best    | 0.673         | 0.016     |
> |            | Implicit       | 0.673         | 0.009     |
> |            | All Features   | 0.674         | 0.012     |
> | GPT‑4o     | ACTMED         | 0.730         | 0.010     |
> |            | Random         | 0.643         | 0.024     |
> |            | Global Best    | 0.714         | 0.005     |
> |            | Implicit       | 0.694         | 0.016     |
> |            | All Features   | 0.712         | 0.007     |
>
> ---
>
> ## **Novelty of BED with LLMs**
> While Bayesian Experimental Design (BED) and KL-divergence–based selection are well-established tools, ACTMED is, to our knowledge, the first framework to integrate BED with large language models as generative priors for clinical reasoning in a zero-shot setting. Unlike prior reinforcement learning or decision tree–based approaches, which demand extensive task-specific training and retraining to accommodate new diseases or tests, ACTMED uses LLMs to generate plausible, patient-specific test outcomes on demand. This enables rapid, adaptive reasoning across diverse tasks and clinical domains without any fine-tuning or reliance on large, labelled datasets—a capability not demonstrated in prior work. By decoupling diagnostic reasoning from model retraining, ACTMED fills a critical gap: enabling sequential, uncertainty-aware diagnostic decisions in contexts where labelled data is scarce or clinical conditions shift rapidly. To our knowledge, the only related work applies BED with entropy gains to coding task disambiguation (1), but not to medicine, and does not employ KL divergence.
>
> ---
>
> ## **Diagnostic test cost modelling**
> As the reviewer notes, clinical decision-making often involves balancing the diagnostic value of a test against factors such as invasiveness, availability, and cost. In practice, these trade-offs are typically determined by panels of experienced clinicians rather than by automated systems.
>
> In the current study, ACTMED serves solely as a decision-support tool, recommending tests based on their estimated diagnostic benefit alone. For simplicity and comparability, we assumed unit cost for all tests, as the baseline methods used for comparison do not incorporate cost modelling.
>
> Nevertheless, our framework is fully compatible with cost-aware decision-making. Institutions adopting ACTMED could assign test-specific costs, and the model can directly integrate these weights into its test selection procedure.
>
> ---
>
> ## **Clinician involvement**
> The performance of LLMs against humans in diagnostic settings has been evaluated in prior studies suggesting that experienced clinicians still perform better in complex scenarios (2) and this is discussed in the paper. The authors would like to reemphasize that ACTMED is designed as a test selection tool to support clinicians not as a standalone diagnostic tool. To strengthen this, we conducted a qualitative clinician review of ACTMED’s reasoning pathways (see response to reviewer tmUM), which confirmed its alignment with expert reasoning under uncertainty.
>
> ---
>
> ## **Evaluation of LLM-generated samples**
> We appreciate the reviewer’s concerns regarding the quality of LLM-generated samples. We emphasize that the purpose of the sampling procedure is not to exactly predict the most likely test result for a given patient but to draw plausible values under both disease and non-disease assumptions. A well-calibrated test will naturally produce roughly half of its simulated values deviating significantly from the true outcome.
>
> The “percentage in reasonable range” metric is intended to detect hallucinations rather than assess per-sample accuracy. The reasonable range is empirically defined based on observed data, and the LLM generates samples that fall within this range for 99% of cases, confirming that outputs remain realistic. The full distributions produced are also shown in Appendix E.
>
> The “minimum distance” metric will, as noted, tend toward zero if the model can generate sufficiently diverse samples and enough draws are taken (Appendix E). Since ACTMED is constrained to 10 draws per query, this metric serves primarily to ensure that the number of samples drawn for the experiment is large enough for enabling reliable BED computations.
>
> Addressing the reviewer’s suggestion, we conducted a distribution-level evaluation of generated versus true samples using Wasserstein and Energy distances (see Table 2). We have also updated Figures 3 and 4 to represent these distribution metrics. The manuscript text has been revised for clarity to reflect these results.
>
> **Table 2:** *Normalized Distribution Matching Metrics (Lower is Better)*
>
> | Dataset   | Model        | Avg. Wasserstein (± SD) | Avg. Energy Distance (± SD) |
> |-----------|--------------|--------------------------|------------------------------|
> | Diabetes  | GPT‑4o       | 0.110 ± 0.038           | 0.256 ± 0.079               |
> | Diabetes  | GPT‑4o‑mini  | 0.117 ± 0.046           | 0.269 ± 0.094               |
> | Hepatitis | GPT‑4o       | 0.130 ± 0.157           | 0.265 ± 0.191               |
> | Hepatitis | GPT‑4o‑mini  | 0.173 ± 0.237           | 0.330 ± 0.254               |
> | Kidney    | GPT‑4o       | 0.082 ± 0.055           | 0.203 ± 0.102               |
> | Kidney    | GPT‑4o‑mini  | 0.082 ± 0.057           | 0.203 ± 0.104               |
>
>
> ---
>
> ## **Interpretation of model results and stoppage criterion**
> CKD results, originally included only in the appendix due to near-perfect performance across all methods, are now presented in Figure 5 for completeness. Figure 7 examines why the implicit baseline underperforms relative to ACTMED. For datasets where ACTMED shows clear performance gains (diabetes and hepatitis), we further analyzed feature selections relative to the globally optimal sets identified prior to observing any data.
>
> ACTMED’s stopping criterion is not based on the model’s state but is determined by the KL divergence between successive belief states. For example, with γ = 0.5, the framework will acquire another test only if the expected shift in belief is at least half the distance to the decision boundary. If this threshold is not met, the system deems itself sufficiently confident that no additional test would meaningfully alter the prediction and halts testing.
>
> Clinical experts involved in the review noted that this rule closely mirrors their own decision-making in practice, where additional testing is pursued only when it is likely to significantly change diagnostic confidence.
>
> ---
>
> ## **Comparison of ACTMED and full information baseline**
> We agree with the reviewer that ACTMED outperforming the full features baseline, which is not limited to 3 diagnostic tests but has access to all information in the dataset may seem surprising. We have conducted an experiment involving classification using sequential increases in the number of features to support this (see response to reviewer pTAQ).
>
> ---
>
> ## **Formatting and minor comments**
> We standardized mathematical notation (per Reviewer pTAQ’s suggestions) and clarified figure content. Figure 1 now clearly indicates that the distributions shown represent prior and posterior disease probabilities. The green box notation was simplified for clarity. We clarified that “ML” and “DL” in Table 3 refer to “machine learning” and “deep learning,” with examples, and corrected the note on the diabetes dataset: we do not claim BMI is diagnostic, only that OGTT is absent, necessitating indirect metrics. The paragraph on line 354 has been rewritten to better explain how ACTMED supports clinicians.
>
> ---
>
> ## **References**
> 1. Active Task Disambiguation with LLMs (Kobalczyk et al., ICLR 2025)
> 2. Evaluation and mitigation of the limitations of large language models in clinical decision-making (Hager et al, Nature Medicine 2024)
>
> ---
>
> *We greatly appreciate the time and effort you’ve dedicated to reviewing our paper. We hope that the resulting additional experiments will strengthen the contributions of this paper. We’d be happy to engage in further discussions.*

---

> ### Comment · Reviewer_fFNv · 2025-08-03
> **Thanks for the points.**
>
> Thanks for the detailed responses. No further questions from my side. I'm willing to raise the score to 5.

---

> > ### Author Response · Authors · 2025-08-05
> >
> > Dear Reviewer,
> >
> > Thank you very much for taking the time to thoughtfully consider our responses. We greatly appreciate your updated score and your recognition of the contributions and improvements made in the revised submission.
> >
> > We’re grateful for your continued engagement throughout the rebuttal process.
> >
> > Best wishes,
> > The authors of submission 22995

---

### Official Review · Reviewer_2RXP · 2025-07-03

**Clarity:** 3
**Significance:** 2
**Originality:** 2
**Rating:** 5
**Confidence:** 2

**Summary:**

The work tried to balance the medical cost and diagnostic accuracy in clinical diagnosis tasks. Based on previous work showing LLM gives very accurate prediction on diagnostics, the authors propose a new pipeline based on greedy selection of test with most information gain with the help of LLM estimating the prior and posterior distribution. Solid experimental studies were conducted to verify the effectiveness of the proposed approach.

**Questions:**

See weakness

**Ethical Concerns:**

["NO or VERY MINOR ethics concerns only"]

**Quality:**

3

**Strengths And Weaknesses:**

- Strength

The methodology itself is quite intuitive and the paper is overally well presented.

- Weakness

Given that the quality of proposed approach heavily relies on the fact that LLM gives accuracy prediction on diagnostic, is there any quantitative characterization of possible error rate if LLM has some possibility to give wrong predictions? The reviewer believe if such error rate can be well controlled than the proposed method can be more convincing.

---

> ### Author Rebuttal · Authors · 2025-07-30
>
> *We thank the reviewer for their positive assessment of our work and for recognizing both the intuitive methodology and overall presentation. Below, we address the specific concern regarding potential LLM prediction errors.*
>
> ---
>
> ## **Quantifying LLM Uncertainty and Error Rates**
> We agree that the reliability of the framework depends on the accuracy of the LLM-generated diagnostic predictions. To address this, we now provide two complementary evaluations:
>
> ### 1.Repeated sampling and bootstrapping:
> For each patient, we perform 10 independent diagnostic for risk classification and report the mean prediction accuracy with 95% confidence intervals using Bayesian bootstrapping (1). This approach captures the variability of model outputs and allows us to estimate worst-case bounds on prediction accuracy. Table 1 shows the mean, standard deviation, and Bayesian bootstrap confidence intervals for both models across the three datasets with patients stratified by true disease label. Table 2 summarizes the metrics across the entire dataset. A more detailed discussion is included in the Appendix E.
>
> **Table 1:** *Average model prediction accuracy across the hepatitis, diabetes, and CKD datasets (positive and negative cases, both models), with corresponding 95% confidence intervals.*
>
> | Dataset   | Model        | Label | Mean Avg. Risk | Std. Avg. Risk | CI Lower | CI Upper |
> |-----------|--------------|-------|----------------|----------------|----------|----------|
> | Diabetes  | GPT‑4o       | 0     | 0.548          | 0.275          | 0.512    | 0.583    |
> |           | GPT‑4o       | 1     | 0.799          | 0.143          | 0.782    | 0.815    |
> |           | GPT‑4o‑mini  | 0     | 0.716          | 0.223          | 0.686    | 0.744    |
> |           | GPT‑4o‑mini  | 1     | 0.856          | 0.039          | 0.836    | 0.875    |
> | Hepatitis | GPT‑4o       | 0     | 0.093          | 0.129          | 0.081    | 0.106    |
> |           | GPT‑4o       | 1     | 0.561          | 0.320          | 0.539    | 0.583    |
> |           | GPT‑4o‑mini  | 0     | 0.167          | 0.147          | 0.157    | 0.178    |
> |           | GPT‑4o‑mini  | 1     | 0.581          | 0.289          | 0.555    | 0.605    |
> | Kidney    | GPT‑4o       | 0     | 0.039          | 0.033          | 0.029    | 0.051    |
> |           | GPT‑4o       | 1     | 0.907          | 0.060          | 0.898    | 0.916    |
> |           | GPT‑4o‑mini  | 0     | 0.179          | 0.119          | 0.161    | 0.197    |
> |           | GPT‑4o‑mini  | 1     | 0.908          | 0.061          | 0.897    | 0.917    |
>
> ### 2.Distribution-level validation:
> We compute Wasserstein and Energy distances between the generated and true test distributions (see also response to reviewer fFNv), providing a measure of how well the LLM replicates the observed data beyond point predictions.
> Across all three datasets, ACTMED maintains low distributional distances (Wasserstein ≤ 0.15) and stable performance under repeated sampling. For smaller models that fail to produce realistic distributions, overall accuracy declines and ACTMED does not yield performance gains, as effective Bayesian Experimental Design depends on accurate sampling (see also response to reviewer tmUM).
>
> ---
>
> ## **Strengthening Experimental Validation**
> In addition, we have incorporated several experiments requested by other reviewers, including:
> 1. Evaluations using open-source models.
> 2. Multi-disease diagnostic tasks derived from the AgentClinic dataset.
> 3. Expanded clinician-in-the-loop evaluations.
>
> ---
>
> ## **References**
> 1.	The Bayesian Bootstrap (Donal B. Rubin, The Annals of Statistics, 1981)
>
> ---
>
> *We appreciate the time and effort you have dedicated to reviewing our paper and hope these additions and clarifications address your concerns satisfactorily. We’d be happy to engage in further discussions.*

---

### Official Review · Reviewer_tmUM · 2025-07-21

**Clarity:** 2
**Significance:** 2
**Originality:** 2
**Rating:** 3
**Confidence:** 4

**Summary:**

This paper presents an LLM-based clinical diagnosis framework that actively selects and updates its belief on tests to make decisions on disease states. Experiments were conducted on three different datasets using two GPT models and the results were also compared with various feature selection-based baseline approaches using the same models and risk prediction prompts. Results showed the effectiveness of the proposed methods compared to considered approaches and baselines across various capabilities including timely diagnosis and transparent explanations.

**Questions:**

see above.

**Ethical Concerns:**

["NO or VERY MINOR ethics concerns only"]

**Final Justification:**

I've reviewed the rebuttal responses and other reviewers' comments and author responses that helped with additional clarity and context about the key contributions of the paper, limitations, and some efforts/ongoing work to address those limitations. However, I am still not convinced on the value-add of ACTMED as it requires a strong LLM to perform better, so not sure if the results would be model agnostic as mentioned in the paper. Regardless, based on the new information and results shared by the authors, I've adjusted my rating accordingly.

**Limitations:**

yes

**Quality:**

2

**Strengths And Weaknesses:**

Strengths:

- The paper addresses an important problem of clinical diagnosis using LLMs and a probabilistic framework, where LLM is used as a flexible simulator to predict test outcomes to update the disease belief states while it does not need any task-specific training.

- The proposed approach is agnostic to LLM model architecture and shows promising results within the considered experimental setup and comparisons with various baselines.

- Rigorous experiments have been carried out on 3 datasets demonstrating the generalizability of the proposed methods.

- Extended results with clinician-in-the-loop review process and computational cost analysis have been reported that would be beneficial for the community.


Weaknesses:

- Evaluation is weak given the paper considers only two closed source GPT-based models for experimentation lacking other open-source models, hindering detailed understanding of why the proposed methods work or do not work in certain cases and reproducibility of the experiments.

- The experiments have been conducted in a strictly controlled setting, where the task is formulated as a binary classification task making it far from the real-world clinical diagnosis inference task requiring multi-label categorization and dependencies.

- The paper only considers categorical and numerical features ignoring free text EHR reports that raises questions about the usefulness of using LLMs for this task.

- The related work section seems to be weak as well, as it omits various state-of-the-art approaches using DRL that also mimics similar iterative clinical decision-making process (e.g., https://aclanthology.org/I17-1090/)

- Although clinician-in-the-loop evaluation process was proposed but the paper lacked a thorough qualitative evaluation of the proposed approaches along with detailed examples of how the methods could have been improved further.

- The overall writing, structure, organization and content of the paper can be improved, as at times the proposed objectives, frameworks, experimental setup etc. seemed confusing and not coherent.

---

> ### Author Rebuttal · Authors · 2025-07-30
>
> *We thank the reviewer for their constructive feedback and insightful comments. We appreciate that the reviewer recognizes both the importance and clinical relevance of our work, as well as the detailed evaluation of its performance across three datasets. Below, we address each concern raised.*
>
> ---
>
> ## **Experimental evaluation and model choice**
> As we emphasize in the paper, ACTMED is inherently model-agnostic: it does not rely on any model-specific fine-tuning or internal access, but only on generated distributions of test outcomes and disease risk estimates. Our contribution is a test-time computational improvement — selecting tests to maximize information gain — rather than a modification of the underlying language model. Consequently, the primary consideration is not whether the model is open-source or closed-source, but whether it is capable of generating sufficiently accurate and diverse outcome distributions.
>
> To clarify this point, we reran all experiments using Biomistral-7B, an open-source biomedical LLM. The results (Tables 1 and 2) show that while ACTMED is compatible with Biomistral, its diagnostic performance depends heavily on the quality of the generated distributions. Biomistral’s outputs frequently diverged from real data — for example, predicting non-integer values for the number of prior pregnancies in the diabetes dataset — leading to weaker downstream performance.
>
> **Table 1:** Wasserstein distance between the true empirical distributions and the model-generated test outcome distributions.
> | Dataset   | Model        | Avg. Wasserstein (± SD) |
> |-----------|--------------|--------------------------|
> | Diabetes  | GPT‑4o       | 0.110 ± 0.038           |
> | Diabetes  | GPT‑4o‑mini  | 0.117 ± 0.046           |
> | Diabetes  | Biomistral      | 0.125 ± 0.048           |
> | Hepatitis | GPT‑4o       | 0.130 ± 0.157           |
> | Hepatitis | GPT‑4o‑mini  | 0.173 ± 0.237           |
> | Hepatitis | Biomistral      | 0.425 ± 0.453           |
> | Kidney    | GPT‑4o       | 0.082 ± 0.055           |
> | Kidney    | GPT‑4o‑mini  | 0.082 ± 0.057           |
> | Kidney    | Biomistral      | 0.215 ± 0.225           |
>
> Lower values indicate that the generated samples more closely reflect the real-world data. GPT models produce distributions that closely match the data, while Biomistral-7B shows substantially higher distances, suggesting its generated outcomes are less realistic.
>
> **Table 2:** Classification accuracy using all features versus ACTMED-selected and sequential implicit-selected features for each model.
> | Dataset   | Model       | ACTMED (Mean ± Std)   | All Features (Mean ± Std) | Implicit (Mean ± Std)     |
> |-----------|-------------|------------------------|----------------------------|----------------------------|
> | Diabetes  | GPT-4o      | 0.682 ± 0.012         | 0.584 ± 0.011              | 0.544 ± 0.004              |
> |           | GPT-4o-mini | 0.593 ± 0.039         | 0.451 ± 0.062              | 0.488 ± 0.040              |
> |           | Biomistral     | 0.529 ± 0.025         | 0.450 ± 0.008              | 0.554 ± 0.034              |
> | Kidney    | GPT-4o      | 0.999 ± 0.003         | 1.000 ± 0.000              | 0.999 ± 0.003              |
> |           | GPT-4o-mini | 0.992 ± 0.006         | 0.975 ± 0.006              | 0.997 ± 0.005              |
> |           | Biomistral     | 0.657 ± 0.021         | 0.564 ± 0.033              | 0.620 ± 0.045              |
> | Hepatitis | GPT-4o      | 0.839 ± 0.013         | 0.807 ± 0.004              | 0.729 ± 0.016              |
> |           | GPT-4o-mini | 0.825 ± 0.013         | 0.793 ± 0.009              | 0.770 ± 0.007              |
> |           | Biomistral     | 0.489 ± 0.029         | 0.511 ± 0.027              | 0.501 ± 0.042              |
>
> ACTMED improves diagnostic accuracy when paired with models that generate realistic outcomes (low distances in Table 1). For models like Biomistral-7B, where outputs diverge significantly from the data, ACTMED cannot provide meaningful gains.
>
> ---
>
> ## **Binary vs. Multi-Label Diagnosis**
> Clinical diagnosis generally follows a two-step process (1). First, the clinician conducts an initial assessment, forming a primary diagnosis alongside a list of differential diagnoses. Second, tests are ordered to both confirm the leading diagnosis and rule out competing conditions.
>
> This process naturally reduces to a series of one-vs-all evaluations, where each condition is assessed independently. Such an approach is common in clinical practice and machine learning when handling complex multi-class problems. Widely used clinical scores, such as the Wells score for deep vein thrombosis (2) and the CHA₂DS₂ VASc score for atrial fibrillation-related stroke (3), each estimate the likelihood of a single disease. Clinicians routinely apply these sequentially, allowing patients to be assessed for multiple conditions in parallel.
>
> ACTMED addresses the second step by optimizing test selection across diseases, independent of how the differentials are generated. This distinction has been clarified in the revised manuscript, and we include results on a multi-disease dataset derived from AgentClinic (see response to reviewer fFNv), where ACTMED continues to perform strongly.
>
> ---
>
> ## **Handling of Categorical and Free-Text Features**
> Numerical and categorical variables are converted into natural language descriptions resembling EHR notes (see Appendix D). This ensures that values are presented in a clinically interpretable way, accounting for differences in units and context. At the same time, ACTMED relies on a tabular structure so each feature can be sampled and recombined consistently, allowing us to dynamically generate clinical vignettes and counterfactual diagnostic pathways for the same patient — something not feasible with free-text records alone.
>
> To address the reviewer’s concern, we added an experiment using a dataset derived from AgentClinic that combines structured features with free-text EHR-style notes. ACTMED operates effectively in this mixed-modality setting without modification, demonstrating its flexibility.
>
> ---
>
> ## **Comparison with DRL-Based Approaches**
> We thank the reviewer for pointing out additional related work. In the revised manuscript, we now cite Ling et al. (4) along with related DRL-based clinical decision-making systems by Yu et al. (5) and Zhong et al. (6). These works demonstrate how reinforcement learning can iteratively guide test selection and decision-making in healthcare, and we view them as complementary to our approach.
>
> ACTMED differs in that it combines Bayesian Experimental Design with LLM-based generative priors to simulate plausible test outcomes at inference time. This allows ACTMED to operate without task-specific retraining, whereas DRL-based methods typically require substantial training to adapt to new tasks.
>
> ---
>
> ## **Clinician-in-the-Loop Evaluation**
> We conducted a clinician-in-the-loop evaluation with three experienced clinicians and two senior medical students, evaluating 450 test decisions (10 traces across 3 datasets, each assessed by 5 evaluators with 3 decisions per trace). Experts judged ACTMED’s test selections and the resulting risk adjustments as clinically reasonable in 94.5 ± 1.4 % of cases, underscoring its reliability as a decision-support tool.
>
> Clinicians emphasized that they would be reluctant to trust diagnostic suggestions from a black-box LLM without transparent reasoning. ACTMED’s step-by-step decision process was repeatedly praised for building confidence in the model reasoning. Several clinicians noted that ACTMED’s Bayesian decision framework aligns with the reasoning they themselves use when selecting tests under uncertainty, particularly in the absence of clear guidelines.
>
> They also pointed out cases where clinical decision-making diverges from a purely Bayesian approach — for example, when clear guidelines mandate a specific test (such as PCR for COVID 19 diagnosis) or when certain tests are routinely performed despite limited informativeness, such as screening to rule out cancer. However, clinicians agreed that decision support is only needed for the less straightforward cases, where test selection involves genuine uncertainty and no clear protocol exists.
>
> ---
>
> ## **Writing and Presentation**
> In response to Reviewer pTAQ and others, we have significantly improved the manuscript’s clarity and presentation. Sections 2 and 3 were rewritten with clearer, standardized notation, while Figures 1 and 2 were clarified for readability. Section 4.1 now reports Wasserstein and Energy distances to better characterize the generated sample distributions, addressing requests for more rigorous evaluation. Additional reviewer suggestions have also been incorporated throughout, resulting in a clearer, more consistent, and more informative presentation.
>
> ---
>
> ## **References**
> 1. Improving Diagnosis in Health Care (Committee on Diagnostic Error in Health Care National Academies Press (US) 2015)
> 2. Derivation of a simple clinical model to categorize patients probability of pulmonary embolism: increasing the model’s utility with the SimpliRED D-dimer (Wells et al. Thrombosis and haemostasis 2000)
> 3. Validation of risk stratification schemes for predicting stroke and thromboembolism in patients with atrial fibrillation: nationwide cohort study (Olesen et al. BMJ 2011)
> 4. Learning to Diagnose: Assimilating Clinical Narratives using Deep Reinforcement Learning (Ling et al., IJCNLP 2017)
> 5. Deep Reinforcement Learning for Cost-Effective Medical Diagnosis (Yu et al. ICLR 2023)
> 6. Hierarchical reinforcement learning for automatic disease diagnosis (Zhong et al. Bioinformatics 2022)
>
> ---
>
> *Thank you for your thoughtful feedback. We appreciate your insights, which have helped improve the quality and clarity of our work. We believe ACTMED provides a practical step toward generalizable, interpretable AI in clinical diagnostics, and we welcome continued dialogue.*

---

> ### Comment · Reviewer_tmUM · 2025-08-01
>
> Thanks for the detailed responses, very helpful. Few comments:
>
> - For the evaluation and model choice aspect, it would be useful to also consider comparisons with some open-source general purpose models (e.g., Llama) to further clarify the key contributions of the paper based on a diverse set of models, because in its current setting it's still somewhat difficult to understand the usefulness of the proposed methods based on models from two very opposite dimensions (closed source general purpose vs. open source domain-specific models).
>
> - Also, as you acknowledged based on other reviewers' comments that "the performance of ACTMED depends on the underlying LLM’s ability to generate plausible test outcomes.", so I am not quite sure how to interpret the "ACTMED is inherently model-agnostic" claim.

---

> > ### Author Response · Authors · 2025-08-04
> >
> > *Thank you for your thoughtful evaluation of our rebuttal. We appreciate your continued engagement and the valuable suggestions for clarification.*
> >
> > ---
> >
> > The central contribution of this work is to demonstrate the potential of LLMs as probabilistic simulators for decision-making in clinical test selection. We would like to clarify that we do not claim ACTMED improves the diagnostic performance of every LLM. Rather, when we state that ACTMED is model-agnostic, we mean that the method does not rely on any model-specific architecture, fine-tuning, or internal access. In principle, it can be paired with any language model capable of generating outcome and risk distributions at test time.
> >
> > However, as with all Bayesian Experimental Design (BED) frameworks, the quality of test selection inherently depends on the accuracy and variability of the underlying simulator. In this context, the generative ability of the LLM to sample plausible test outcomes under different disease states is critical.
> >
> > ---
> >
> > To further evaluate ACTMED's requirements and limitations, we conducted new experiments using the LLaMA 3.3-70B model. The model performs well when all features are available and is generally capable of sound diagnostic reasoning (see Table 1). However, it struggles to generate diverse and clinically coherent test outcome distributions required for ACTMED. Across multiple prompts and temperature settings (including >1.0), the model almost exclusively produced identical outputs across all 10 samples, suggesting a failure to capture conditional variability under disease presence versus absence — a core requirement for ACTMED’s information-theoretic reasoning. This renders ACTMED inoperable in such settings.
> >
> >
> > **Table 1.** *Accuracy of different models on the Hepatitis, Diabetes, and Chronic Kidney Disease datasets using all features and ACTMED-selected features.*
> >
> > | Dataset   | Model         | ACTMED (Mean ± Std) | All Features (Mean ± Std) |
> > |-----------|---------------|---------------------|----------------------------|
> > | Diabetes  | GPT-4o        | 0.682 ± 0.012       | 0.584 ± 0.011              |
> > |           | GPT-4o-mini   | 0.593 ± 0.039       | 0.451 ± 0.062              |
> > |           | Biomistral    | 0.529 ± 0.025       | 0.450 ± 0.008              |
> > |           | LLaMA 70B     | 0.540 ± 0.013       | 0.474 ± 0.012              |
> > | Kidney    | GPT-4o        | 0.999 ± 0.003       | 1.000 ± 0.000              |
> > |           | GPT-4o-mini   | 0.992 ± 0.006       | 0.975 ± 0.006              |
> > |           | Biomistral    | 0.657 ± 0.021       | 0.564 ± 0.033              |
> > |           | LLaMA 70B     | 0.984 ± 0.003       | 0.986 ± 0.000              |
> > | Hepatitis | GPT-4o        | 0.839 ± 0.013       | 0.807 ± 0.004              |
> > |           | GPT-4o-mini   | 0.825 ± 0.013       | 0.793 ± 0.009              |
> > |           | Biomistral    | 0.489 ± 0.029       | 0.511 ± 0.027              |
> > |           | LLaMA 70B     | 0.699 ± 0.023       | 0.793 ± 0.010              |
> >
> >
> >
> >
> >
> >
> > We acknowledge this as a limitation in the revised manuscript: ACTMED currently requires strong generative capabilities that are primarily available in the most advanced LLMs. However, we believe this is a temporary constraint. The rapid pace of progress in both large and smaller models suggests that broader compatibility is likely to be achievable in the near future.
> >
> > ---
> >
> > *We hope this clarifies the model-agnostic nature of our framework and the practical considerations when using ACTMED with different LLMs. Thank you again for your valuable feedback.*

---

> > > ### Comment · Reviewer_tmUM · 2025-08-06
> > >
> > > Thank you for the detailed response, very helpful.

---

> > > > ### Author Response · Authors · 2025-08-09
> > > >
> > > > Dear Reviewer,
> > > >
> > > > Thank you very much for taking the time to thoughtfully consider our responses and for your constructive engagement throughout the process. We greatly appreciate your recognition of the contributions and improvements in the revised submission.
> > > >
> > > > Best wishes,
> > > >
> > > > The authors of submission 22995

---

### Decision · Program_Chairs · 2025-09-17

**Decision:**

Accept (poster)

**Comment:**

The paper introduces a diagnostic framework that integrates (Bayesian) experimental design with LLMs to guide adaptive clinical test selection; its goal is to improve accuracy, interpretability, and cost-effectiveness across datasets. Reviewers appreciated the novelty and the application relevance of the approach, as well as the extensive experiments and clinician-in-the-loop evaluations. However, they expressed reservations about the evaluation design, including reliance on closed-source models, simplified binary tasks, omission of free-text EHR features, and questions about whether the approach is truly model-agnostic. Overall sentiment is cautiously positive: reviewers found the idea important and timely, but felt the claims and experiments needed stronger grounding. To strengthen the work, the authors should candidly discuss its limitations—such as dependence on strong LLMs, restricted evaluation settings, and gaps in multimodal or multi-label clinical contexts—so that readers can better understand the scope, generalizability, and future directions of the proposed framework.